# Remodeling of the ribosomal quality control and integrated stress response by viral ubiquitin deconjugases

Jiangnan Liu [1], Noemi Nagy [1], Carlos Ayala-Torres[1], Francisco Aguilar-Alonso [1,2], Francisco Morais-Esteves [1,3], Shanshan Xu[1] & Maria G. Masucci [1]✉

The strategies adopted by viruses to reprogram the translation and protein quality control machinery and promote infection are poorly understood. Here, we report that the viral ubiquitin deconjugase (vDUB)−encoded in the large tegument protein of Epstein-Barr virus (EBV BPLF1)−regulates the ribosomal quality control (RQC) and integrated stress responses (ISR). The vDUB participates in protein complexes that include the RQC ubiquitin ligases ZNF598 and LTN1. Upon ribosomal stalling, the vDUB counteracts the ubiquitination of the 40 S particle and inhibits the degradation of translation-stalled polypeptides by the proteasome. Impairment of the RQC correlates with the read-through of stall-inducing mRNAs and with activation of a GCN2-dependent ISR that redirects translation towards upstream open reading frames (uORFs)- and internal ribosome entry sites (IRES)-containing transcripts. Physiological levels of active BPLF1 promote the translation of the EBV Nuclear Antigen (EBNA)1 mRNA in productively infected cells and enhance the release of progeny virus, pointing to a pivotal role of the vDUB in the translation reprogramming that enables efficient virus production.

Viruses strictly depend on the host cell translation machinery for efficient protein synthesis. They have thus evolved multiple mechanisms to ensure that the translation of viral mRNA occurs at levels above and beyond those of cellular mRNA[1]. This is remarkable since the translation of viral mRNAs is likely to be challenging due to the presence of features that are known to slow translation and induce ribosome stalling, such as long repetitive sequences, complex secondary structures, nucleotide misincorporations and suboptimal codon usage[2,3]. While pausing the elongation cycle may resolve some of the issues associated with the translations of challenging mRNAs[4], persistent stalling will cause ribosome collision that may disrupt protein synthesis and ultimately cause cell death[5].

To rescue stalled ribosomes, the cell deploys a ribosome-associated quality control (RQC) machinery that recognizes collided di- and trisomes[6,7] and triggers site-specific ubiquitination of the 40 S particle[8–11]. In mammals, the ubiquitination of RPS10 (eS10) and RPS20 (uS10) by ZNF598[8,9,12–15], promotes splitting of the 80 S ribosome by the ASCC complex[7,16], followed by ubiquitination of the peptidyl-tRNA associated with the 60 S ribosome by Listerin-1 (LTN1), extraction of the ubiquitinated substrate by the p97/VCP AAA+ ATPase[17] and degradation by the proteasome[18,19], which allows recycling of the ribosome particle. Although potentially deadly, failure of the RQC may also promote translation because a prolonged pause facilitates the restart of translation downstream of the stall-inducing mRNA sequence[8,14]. Ubiquitination of the 40 S particle is also triggered by mRNA features that stall the ribosomes during the scanning step or at the start codon. This initiation-associated RQC involves the ubiquitination of RPS2 (uS5) and RPS3 (uS3) by the RNF10 ligase[20,21],

[1]Department of Cell and Molecular Biology, Karolinska Institutet, Stockholm, Sweden. [2]Present address: Unidad de Desarrollo e Investigación en Bioterapéuticos (UDIBI), Escuela Nacional de Ciencias Biológicas, Instituto Politécnico Nacional, Mexico City, Mexico. [3]Present address: Department of Viroscience, Erasmus Medical Center, Rotterdam, The Netherlands. ✉e-mail: maria.masucci@ki.se

and was shown to target irreversibly stalled ribosomes for degradation[20–23].

In addition to triggering the RQC, ribosome stalling has also been linked to the initiation of the integrated stress responses (ISR) via activation of the GCN2 kinase[24–26]. This is mediated by the Mitogen-Activated Protein-3-Kinases (MAP3K) Sterile alpha motif and leucine zipper containing kinase ZAKα that is activated upon interaction with collided ribosomes and phosphorylates, in addition to GCN2, the MAPK p38, and c-JUN N terminal kinase (JNK) that mediate a pro-apoptotic Ribosome Stress Response (RSR)[5,25]. The ISR is a cellular resilience response initiated by stress-activated kinases that converge on eukaryotic initiation factor 2α (eIF2α) phosphorylation. Phosphorylation converts eIF2α from a substrate to a competitive inhibitor of the guanine exchange factor eIF2B[27], resulting in depletion of the ternary initiator complex and consequent global inhibition of protein synthesis. This alleviates the burden of potentially proteotoxic products while allowing the translation of mRNAs that encode proteins required for cell survival and recovery, such as the transcription factor ATF4 and several antiapoptotic proteins[28]. This translational reprogramming occurs at the level of translation initiation and involves the recognition of structural features found in many cellular and viral mRNAs[29], which allows translation restart at upstream open reading frames (uORFs)[30–32], or the cap-independent recruitment of the 43 S particle to internal ribosomal entry sites (IRES)[33,34].

How the cell coordinates the activation of RQC and ISR in response to ribosomal and proteotoxic stress is still poorly understood, but compounding evidence points to their interdependence since the inactivation of one pathway was found to correlate with the overactivation of the other[26,35]. Both pathways were shown to require ubiquitination of the 40 S ribosome, but they appear to engage different ribosomal proteins. While the ZNF598-mediated ubiquitination of RPS10 and RPS20 initiates the RQC, activation of the ISR is accompanied by ubiquitination of RPS2 and RPS3 by RNF10[21,36]. These ubiquitination events were shown to be hierarchically organized and reversed by deubiquitinases, including USP21, OTUD3, and USP10[20–23], that regulate the intensity and duration of the stress responses.

The challenging structure of many viral mRNAs and the need to mass produce comparatively small proteomes suggest that viruses may be exquisitely dependent on the coordinated regulation of RQC and ISR for efficient viral protein synthesis. We have explored this issue in the context of infection by Epstein-Barr virus (EBV), a lymphotropic herpesvirus that establishes persistent infection in most human adults and participates in the pathogenesis of lymphoid and epithelial cell malignancies[37]. Increased levels of circulating virus and antibodies to structural viral proteins precede the clinical manifestation of EBV-associated malignancies and non-malignant autoimmune disorders[38–40], pointing to an important role of the productive virus cycle in disease pathogenesis. During productive EBV infection, more than seventy viral proteins reprogram the cellular environment to allow virus replication and provide the building blocks of new virus particles[41]. The known viral mechanisms for interfering with translation often involve the expression of RNA-binding proteins that hijack, mimic, or inactivate components of the translation machinery[1]. Here, we outline a strategy by which EBV may reprogram translation by harnessing the activity of the ubiquitin deconjugase encoded in the N-terminal domain of the large tegument protein BPLF1 (vDUB) to inhibit the RQC and trigger a GCN2 kinase-dependent ISR. Our findings suggest that the coordinated regulation of these pathways by the vDUB may greatly enhance the translation of viral mRNAs during productive infection.

## Results

### BPLF1 participates in protein complexes involved in translation and ribosomal quality control

The Gene Ontology (GO) classification of proteins co-immunoprecipitated by the vDUB produced by caspase-1 cleavage of the N-terminus (aa 1-235)

of the EBV large tegument protein BPLF1[42] identified a broad range of cellular functions underlying a pleiotropic role in host-cell remodeling[43]. Upon reanalysis of the mass spectrometry data (Supplementary Data 1), we noticed that approximately half of the interacting proteins are annotated in GO classes that regulate RNA metabolism, ribosome biogenesis, mRNA translation, and proteotoxic cell responses (Fig. 1a). STRING analysis of the protein interaction network identified a major hub centered around the ribosome and components of the mRNA translation, ER trafficking, and ER stress response machinery (Fig. 1b and Supplementary Data 2). The enriched proteins included the translation pre-initiation complex subunit eIF2α that, upon phosphorylation, becomes an inhibitor of global protein synthesis[44], the translation initiation complex subunits eIF4G1, eIF4G2[45] and eIF4E2[46], and the ribosome pause release factor eIF5A that was recently shown to participate in the eukaryotic RQC[47,48]; several components of the 40 S and 60 S ribosome particles; the SRP68 subunit of the Signal Recognition Particle (SRP) and the β subunit of the SRP receptor (SRPRB)[49]; the SEC63 subunit of the ER translocon[50], and the Ribophorin (RPN)−1 subunit of the translocon-associated N-oligosaccharyltransferase (OST) complex[51]. The interaction network also included the RQC ligase ZNF598 and the AAA+ ATPase VCP/p97 (Fig. 1b).

The interactions were validated by co-immunoprecipitation in cell lysates of HEK293T cells transfected with FLAG-BPLF1, the catalytic mutant FLAG-BPLF1[C61A], or FLAG-empty vector (EV). ZNF598 was readily detected in the BPLF1 immunoprecipitates, independent of enzymatic activity (Fig. 1c), and conversely, BPLF1 was found in the ZNF598 immunoprecipitates. While the interaction between BPLF1 and ZNF598 may not be direct, the findings confirm that the vDUB is recruited to protein complexes containing the ligase that initiates the RQC pathways. This was further substantiated by probing the FLAG immunoprecipitates with a selection of antibodies specific to other members of the protein interaction network, which confirmed the presence of BPLF1 in protein complexes involved in translation and the ER translocation and glycosylation of nascent polypeptides (Fig. 1d). Although they did not meet the mass spectrometry significance threshold, the RQC ubiquitin ligase LTN1, the ER translocon subunit SEC61B, and the ZAKα kinase that is auto phosphorylated on collided ribosomes, were also detected in the immunoprecipitates (Fig. 1d).

### BPLF1 inhibits the activation of RQC and stabilizes RQC substrates

Upon sensing collided ribosomes, ZNF598 initiates the RQC by ubiquitinating RPS10 and RPS20[20,21]. To probe the effect of the vDUB on this process, we first tested whether it affects the ubiquitination of the 40 S particle in cells treated with low doses of the translation elongation inhibitor Anisomycin (ANS) that were previously shown to induce random ribosome stalling and collision[13] (Supplementary Fig. 1). Control and ANS-treated FLAG-empty vector (EV)/BPLF1/BPLF1[C61A] transfected HEK293T cells were lysed in a buffer containing NEM and iodoacetamide to inhibit DUB activity and western blots were probed with antibodies to RPS10 and RPS20. Band shifts corresponding to the size of mono-ubiquitinated RPS10 and RPS20 were detected upon long exposure (L) in the western blots of ANS-treated cells (Fig. 2a). In agreement with reports suggesting that ZNF598 promotes RPS20 polyubiquitination[15], bands of size corresponding to RPS20 mono- and di-ubiquitination were detected by the RPS20-specific antibody. The weakness of the ubiquitinated bands relative to the total amount of RPS10 and RPS20 is in line with the notion that low doses of ANS induce ribosome stalling and collision in a fraction of the ribosomes and that ubiquitination is counteracted by the activity of endogenous DUBs. The intensity of the ubiquitinated bands was regularly decreased in cells expressing catalytically active BPLF1, while expression of the catalytic mutant BPLF1[C61A] had no consistent effect (Fig. 2b). To further investigate whether BPLF1 directly affects ZNF598-dependent ubiquitination, a CRISPR/Cas9 knockdown strategy was used to produce a ZNF598 negative subline of HEK293T. In accordance

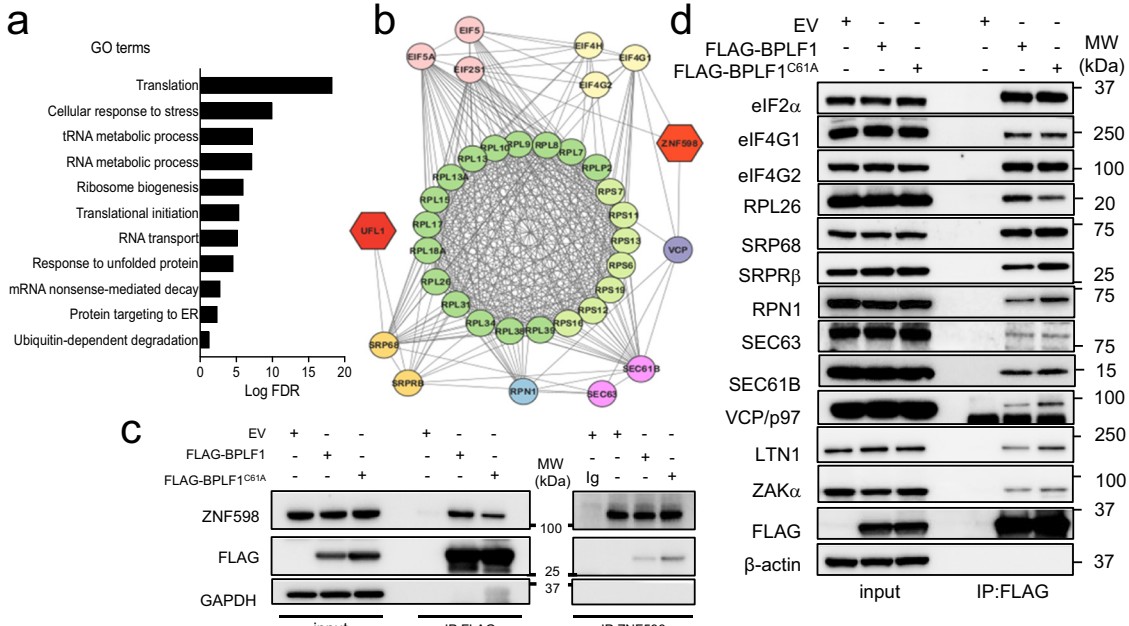

**Fig. 1 | BPLF1 interacts with proteins involved in translation and ribosomal stress responses. a** Enriched biological processes in the BPLF1 interactome identified by Gene Ontology analysis. One hundred sixty-nine of 375 proteins exhibiting fold enrichment ≥2 (45% of the interactome) are annotated in these processes. **b** The STRING network diagram of the major protein interaction hub includes several ribosome subunits, components of the mRNA translation, ER trafficking, and ER stress responses. The functional annotation is color-coded: light green, 40 S ribosome subunit; dark green 60 S ribosome subunit; pink, translation pre-initiation complex; light yellow, translation initiation complex; dark yellow, signal recognition particle (SRP) and SRP receptor (SRPRB) involved in the recognition and translocation of signal-sequence-tagged proteins; magenta, ER translocon complex that mediates forward and retrograde transport across the ER; light blue, translocon-associated N-oligosaccharyltransferase (OST) complex that links high mannose sugars to the Asn-X-Ser/Thr consensus motif of ER-translocated poly-peptides; violet, AAA+ ATPase VCP/p97 that participated in the extraction of ubiquitinated polypeptides from protein complexes; red, ubiquitin and UFM1 ligases. **c** BPLF1 interacts with endogenous ZNF598. Cell lysates of HEK293T cells transfected with plasmids expressing FLAG-BPLF1/BPLF1$^{C61A}$ or the FLAG-empty vector (EV) were immunoprecipitated with either anti-FLAG coated beads or antibodies to ZNF598 followed by capture with Protein G-coated beads. An isotype-matched Ig control was included in the ZNF598 immunoprecipitation to verify specificity. Western blots were probed with the indicated antibodies. Blots from one representative experiment out of two are shown in the figure. **d** Representative western blots illustrating the interaction of BPLF1 with a selection of the proteins identified in (**b**). The lysates and immunoprecipitates were run in multiple gels and parallel membranes were probed with a selection of the indicated antibodies based on the expected size and with the FLAG antibody as loading and transfer control. Blots from one out of two independent co-immunoprecipitation experiments are shown.

with previous reports, treatment with low doses of ANS failed to induce the ubiquitination of RPS10 in HEK-ZNF598-KO cells (Supplementary Fig. 2a). Reconstitution of ZNF598 by transfection of a ZNF598 expressing plasmid resulted in the accumulation of readily detectable ubiquitinated RPS10 and RPS20 species even in the absence of ANS treatment (Fig. 2c). This ubiquitination event was abolished by co-transfecting ZNF598 with the catalytically active BPLF1 (Fig. 2c). In contrast, the catalytic mutant BPLF1$^{C61A}$ had no effect, supporting the conclusion that the vDUB directly counteracts the ubiquitination of 40 S ribosomal proteins by ZNF598.

To test whether subsequent steps of the RQC pathway are affected, we took advantage of a GFP-reporter lacking the termination codon, GFPnonSTOP (Fig. 2d). The progression of translation through the poly(A) tail leads to ribosome stalling, activation of the RQC, ubiquiti-nation of the C-terminally extended product by LTN1, and degradation by the proteasome[52,53]. Western blots of HEK293T cells transfected with the GFPnonSTOP reporter together with FLAG-BPLF1/BPLF1$^{C61A}$, or FLAG-EV with or without treatment with proteasome inhibitor MG132 during the last 6 h before harvesting, were probed with antibodies to GFP. Cells transfected with the parental plasmid were included as a reference for the size of regular GFP. A weak GFP band was detected in cells transfected with the GFPnonSTOP plasmid alone, while treatment with MG132 promoted the accumulation of a slightly larger C-terminally extended product (Fig. 2e). An even more substantial accumulation of the extended product was observed in BPLF1-expressing cells, while the BPLF1$^{C61A}$ mutant had no effect (Fig. 2e, f). In line with the notion that

ZNF598 plays a key role in the initiation of the RQC, the reporter was also stabilized in ZNF598-KO cells, albeit at lower levels compared to MG132 treatment or BPLF1 expression, which is likely explained by the synergistic effect of substrate deubiquitination and proteasome inhibition (Supplementary Fig. 3). These findings support the conclusion that the viral enzyme inhibits the RQC by counteracting the activity of ZNF598, which protects translation-arrested polypeptides from proteasomal degradation.

### BPLF1 promotes the readthrough of stall-inducing mRNAs
Persistent ribosomal stalling, as observed upon functional inactivation of ZNF598 by knockdown, mutation of the 40 S ubiquitination sites, or overexpression of the USP21 deubiquitinase, allows the translation of stall-inducing mRNAs[14,23]. To investigate whether BPLF1 promotes translation across stall-inducing sequences, we utilized a dual fluor-escence reporter that expresses, from a single mRNA, GFP and RFP separated by a linker encoding the FLAG-tagged villin headpiece (VHP) alone (K0) or fused to a stall-inducing stretch of twenty consecutive AAA encoded Lys residues (K20)[54] (Fig. 3a). The linker region is flanked by Ribosomal 2 A skipping sequences (P2A), which allows the assess-ment of translation before and after the linker. The K0 and K20 reporters were co-transfected in HEK293T cells with FLAG-EV/BPLF1/ BPLF1$^{C61A}$ and the intensity of GFP and RFP was measured by FACS. A linear correlation of GFP and RFP fluorescence was observed in cells expressing the K0 reporter independently of expression of the cata-lytically active or mutant BPLF1 (Fig. 3b, upper panels). This allowed

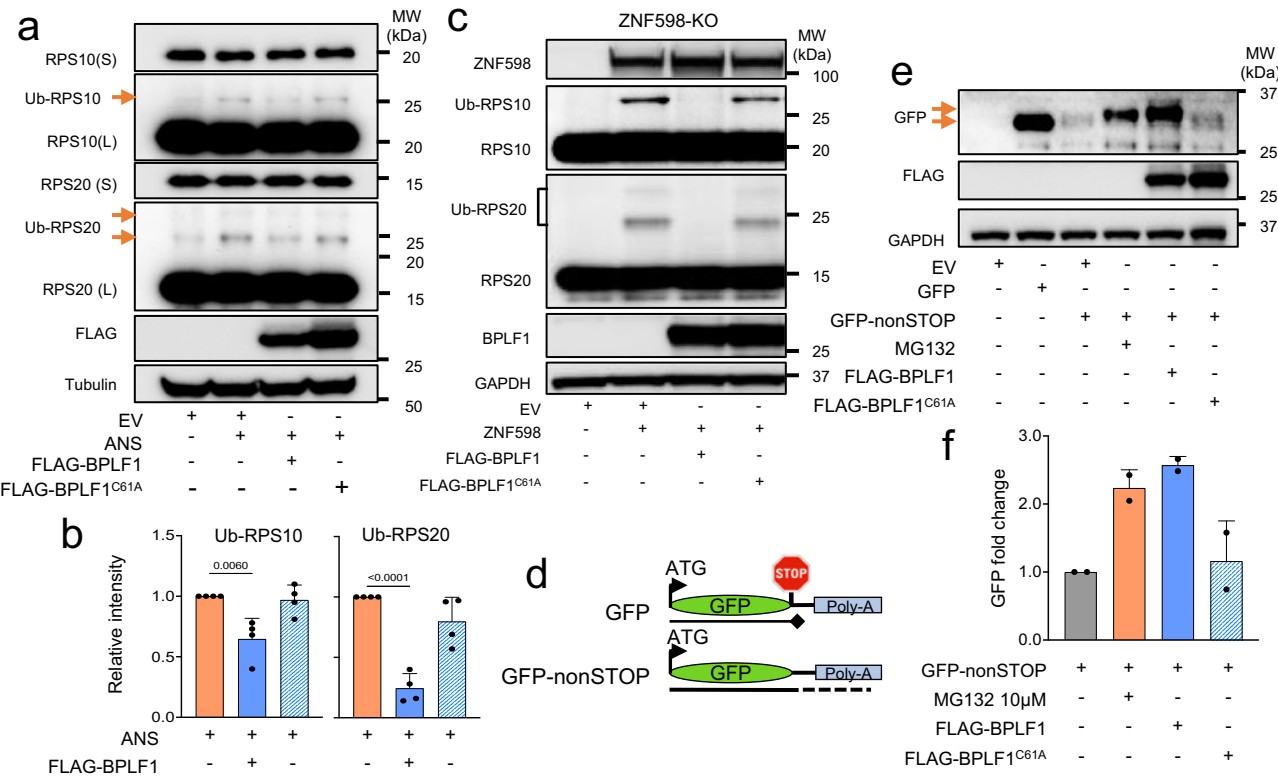

**Fig. 2 | BPLF1 inhibits the activation of RQC in ANS-treated cells and rescues RQC substrates from proteasomal degradation. a** BPLF1 inhibits the ubiquitination of 40 S ribosome subunits in ANS-treated cells. HEK293T cells transfected overnight with plasmids expressing FLAG-EV/BPLF1/BPLF1^C61A were treated with 0.5 μg/ml ANS for 20 min and then lysed in buffer containing NEM and iodoacetamide to inhibit DUB activity. Western blots were probed with the indicated antibodies. Blots from one representative experiment out of four are shown in the figure. **b** The intensities of the Ub-RPS10 and Ub-RPS20 bands were quantified by densitometry in four independent experiments. The mean ± SD relative intensity of the bands in treated versus EV transfected cells is shown. Significance was calculated by paired, two-tailed, Student *t*-test. **c** BPLF1 inhibits the ubiquitination of 40 S ribosome proteins by ZNF598. *ZNF598* knockout HEK293T cells were transfected with a ZNF598 expressing plasmid together with FLAG-EV/BPLF1/BPLF1^C61A. After 24 h, the cells were lysed in a buffer containing NEM and iodoacetamide to inhibit

DUB activity, and western blots were probed with the indicated antibodies. Blots from one representative experiment out of two are shown in the figure. **d** Schematic illustration of the GFPnonSTOP reporter. Following deletion of the stop codon, translation through the poly(A) tail leads to ribosome stalling, activation of the RQC, and degradation of the C-terminally extended product by the proteasomes. **e** Western blots of HEK293T cells transfected with a GFP plasmid or the GFPnonSTOP reporter together with FLAG-EV/BPLF1/BPLF1^C61A were probed with antibodies to GFP. As a control for proteasomal degradation, the EV/GFPnonSTOP transfected cells were treated with 10 μM MG132 during the last 6 h before harvesting. Western blots from one representative experiment out of two are shown in the figure. **f** Mean relative intensity of the GFPnonSTOP bands in MG132-treated and FLAG-BPLF1/BPLF1^C61A transfected cells versus untreated cells in two independent experiments.

the gating of a readthrough region (R) where the transfected cells exhibited comparable levels of green and red fluorescence, corresponding to an average RFP/GFP ratio close to 1. Upon transfection of the K20 reporter, robust repression of RFP fluorescence caused the accumulation of ~80% of the cells into a stalling region (S) where the RFP/GFP fluorescence ratio fell well below 1 (Fig. 3b, lower left panel). A similar proportion of stalled cells was observed upon transfection of the catalytic mutant BPLF1^C61A (Fig. 3b, lower right panel), while expression of the active vDUB rescued significant levels of RFP fluorescence (Fig. 3b, lower middle panel). The results were highly reproducible, with 60–80% of the K20/BPLF1 transfected cells found in the R quadrant versus <20% in control and K20/BPLF1^C61A transfectants (Fig. 3c). The RFP rescue achieved by expression of the vDUB was less efficient compared to that observed in ZNF598-KO cells (Supplementary Fig. 4), which is in line with the expected failure to achieve co-expression of K20 reporter and BPLF1 in all transfected cells. Notably, despite the high frequency of RFP rescue, the intensity of RFP fluorescence remained lower in K20/BPLF1 transfected compared to K0/BPLF1 transfected cells (compare Fig. 3b, upper and lower panels). This was confirmed in western blots probed with antibodies to GFP and RFP, which revealed a significantly weaker RFP band in K20/BPLF1

transfected compared to K0/BPLF1 transfected and K0 control cells (Fig. 3d). A likely explanation for this finding is that, following persistent ribosome stalling due to inhibition of the RQC, translation may restart in different frames, giving rise to truncated or non-fluorescence products, as was observed upon loss of ZNF598[8,14,55]. To test this possibility, we compared the relative intensity of the GFP, RFP, and FLAG-VHP specific bands in K0 and K20 transfected cells (Fig. 3d). Indeed, FLAG-VHPK20, a marker of translation readthrough, was highly expressed in BPLF1 transfected cells, resulting in a FLAG-VHPK20/GFP intensity ratio only slightly lower than FLAG-VHPK0/GFP. In contrast, in line with an imprecise restart of translation downstream of the K20 sequence, the relative RFP/GFP ratio, although increased compared to control and BPLF1^C61A transfected cells, remained well below that observed in K0/BPLF1 transfected cells (Fig. 3e).

## BPLF1 promotes ZAKα phosphorylation and activates a GCN2-dependent Integrated Stress Response (ISR)

Persistent ribosome stalling and collision were shown to trigger the ISR and RSR via activation of GCN2, p38, and JNK by the ZAKα kinase. In line with this scenario, we found that the failure to resolve ribosome stalling via activation of the RQC was accompanied by a time-

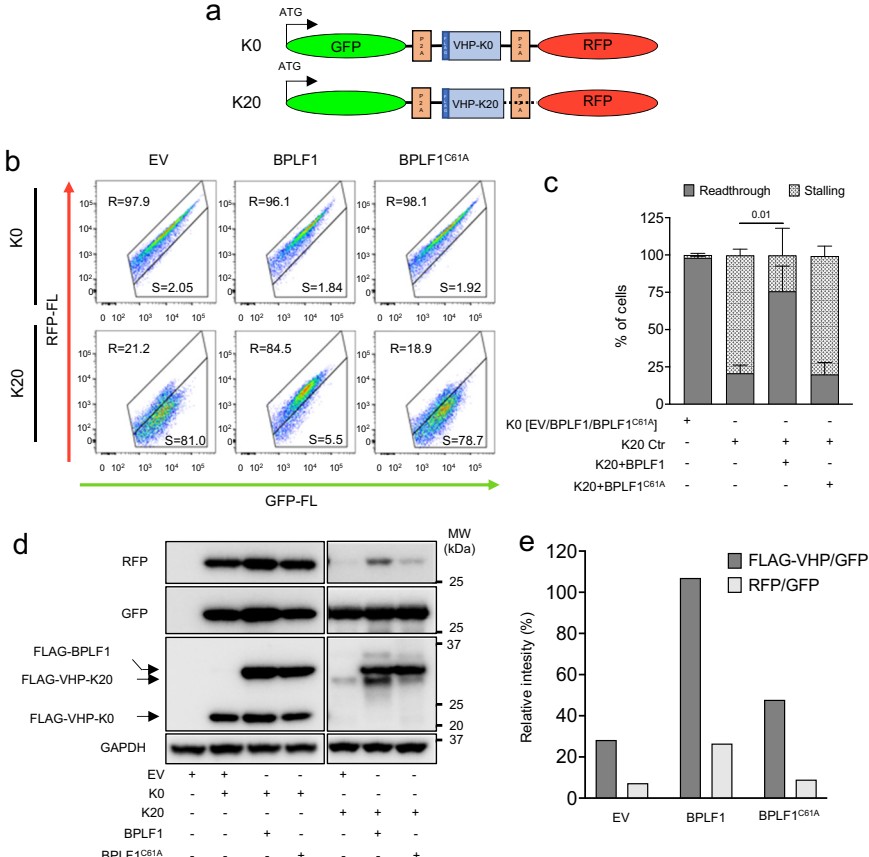

**Fig. 3 | BPLF1 promotes translation elongation across a stall-inducing poly(A) sequence. a** Schematic illustration of the translation-readthrough dual fluorescence reporter. The reporter expresses, from a single mRNA, the GFP and RFP separated by linker encoding a FLAG-tagged villin headpiece (VHP) alone (K0) or fused to a stall-inducing stretch of 20 consecutive lysins residues (K20). The linker regions are flanked by Ribosomal 2 A skipping sequences (P2A), which allows independent assessment of translation before and after the linker. **b** BPLF1 promotes the readthrough of stall-inducing mRNAs. FLAG-EV/BPLF1/BPLF1^C61A transfected HEK293T cells were co-transfected with either the K0 (upper panels) or the K20 reporter (lower panels) before analysis of GFP and RFP fluorescence by FACS. After exclusion of non-fluorescent and dead cells a readthrough (R) region was defined in the plots of K0 transfected cells by gating cells exhibiting linear correlation between GFP and RFP fluorescence. Cells falling in the stalling (S) region exhibited decreased RFP:GFP fluorescence ratios. FACS plots from one

representative experiment out of two are shown in the figure. **c** Mean ± SD of the percentage of transfected cells falling in the (R) and (S) quadrants in three independent experiments. Significance was calculated by paired, two-tailed, Student $t$-test. **d** BPLF1 promotes translation frameshift. Western blots of cell lysates from the experiment shown in Fig. 3b were probed with the indicated antibodies. The GFP, RFP, and FLAG-VHP products were equally expressed in the K0 transfected cells. The RFP and FLAG-VHP-K20 products were virtually undetectable in control cells and cells expressing the BPLF1^C61A mutant. In BPLF1 transfected cells, GFP and FLAG-VHP-K20 were expressed at comparable levels, while RFP was significantly decreased. **e** The intensities of the RFP, GFP, and FLAG-VHP-K0/K20 specific bands in the experiment shown in (**d**) were quantified by densitometry. The data are presented as relative VHP/GFP and RFP/GFP ratios in K20 transfected cells after normalization to the control K0 transfected cells.

dependent increase of ZAKα and eIF2α phosphorylation and accumulation of ATF4 in HEK-ZNF598-KO cells (Supplementary Fig. 2a, b). To investigate whether the expression of BPLF1 may reproduce this phenotype, we first asked whether the active enzyme may promote the phosphorylation of ZAKα, p38 and JNK in ZNF598 proficient cells. Lysates from untreated cells, cells transfected with FLAG-EV/BPLF1/BPLF1^C61A or treated for 30 min with 0.5 μg/ml ANS, were left untreated or treated with alkaline phosphatase for 45 min at 37 °C and western blots were probed with antibodies specific for ZAKα, phospho-p38(Thr180/Tyr182) or phospho-JNK(Thr183/Tyr185). As expected, slow-migrating species corresponding to phosphorylated ZAKα were detected in cells treated with ANS[56] (Fig. 4a). Smaller migration shifts were also detected in EV and BPLF1^C61A transfected cells relative to control and the phosphatase-treated samples, whereas an intermediate shift was observed in cells expressing the active vDUB. The difference in migration shift is consistent with the notion that ANS treatment promotes the auto-phosphorylation of ZAKα at multiple sites that may be differently engaged in response to ribotoxic stressors[56]. Importantly, while a similar migration shift was observed in

cells transfected with FLAG-EV and FLAG-BPLF1^C61A, pointing to a non-specific stress response induced by the transfection procedure, the more substantial shift induced in cells expressing the active vDUB, correlated with enhanced phosphorylation of the ZAKα substrates p38 and JNK (Fig. 4a).

To further assess whether the induction of ZAKα phosphorylation correlates with activation of the ISR, the phosphorylation of eIF2α and the upregulation of ATF4 and its transcriptional target CHOP were investigated in cells transfected with FLAG-BPLF1/BPLF1^C61A. Cells transfected with the FLAG-EV with or without treatment with the ER stress inducer Thapsigargin (TPG)[57] were included as a reference for ISR activation. A comparable upregulation of p-eIF2α, ATF4, and CHOP was observed in TPG-treated and BPLF1-expressing cells, while expression of the BPLF1^C61A mutant had no consistent effect (Fig. 4b, c), supporting the capacity of BPLF1 to activate the ISR. The strong upregulation of ATF4 relative to a comparatively weaker accumulation of p-eIF2α is likely explained by the activation of a feedback loop that, via upregulation of the ATF4 transcriptional target GADD34, promotes the assembly of a PP1-dependent eIF2α-specific phosphatase[58].

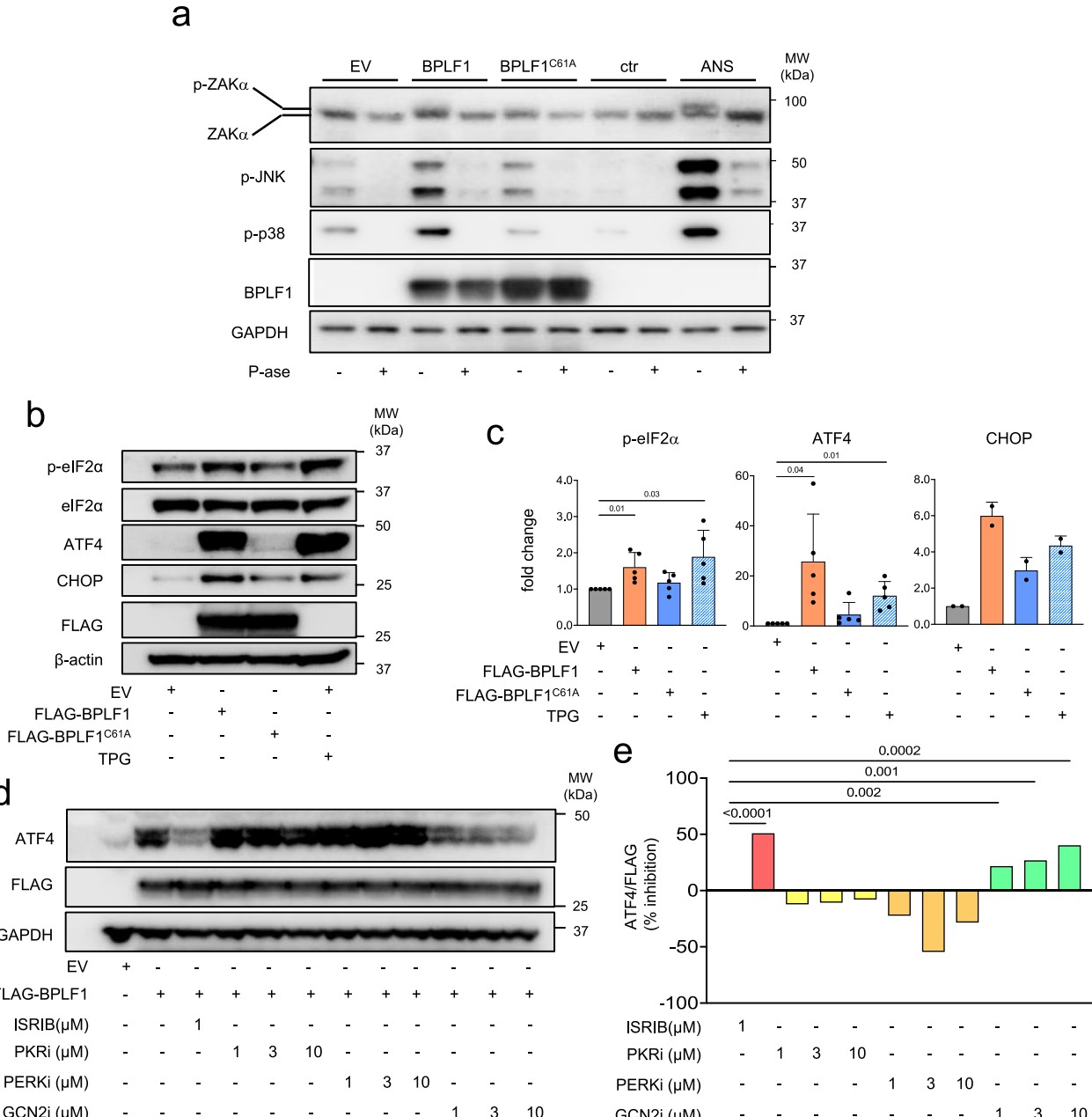

**Fig. 4 | BPLF1 promotes ZAKα phosphorylation and triggers a GCN2-dependent ISR. a** BPLF1 promotes the phosphorylation of ZAKα and the ZAKα substrates p38 and JNK. Lysates of HEK293T cells transfected overnight with FLAG-EV/BPLF1/BPLF1C61A were left untreated or treated for 45 min at 37 °C 1U/μg protein of alkaline phosphatase. Lysates of cells treated with 0.5 μg/ml ANS for 30 min were included as references for ZAKα phosphorylation. Western blots were probed with the indicated ZAKα and phospho-p38 or -JNK antibodies. Western blots from one representative experiment out of three are shown. **b** HeLa cells were transfected with FLAG-EV/BPLF1/BPLF1C61A and cultured overnight. Aliquots of the vector-transfected cells were treated with 1 μM of the ER stress inducer Sarco/endoplasmic reticulum Ca2++ ATPase (SERCA) inhibitor Thapsigargin (TPG) for 2 h before harvesting, and western blots were probed with the indicated antibodies. Blots from one representative experiment out of three are shown in the figure. **c** The intensity of the specific bands was quantified by densitometry in three independent

experiments. The mean ± SD fold change relative to the untreated control is shown. Significance was calculated by paired, two-tailed, Student t-test. **d** Blockade of the ISR prevents the upregulation of ATF4 in BPLF1-expressing cells. BPLF1 transfected HeLa cells were cultured overnight and then treated with the indicated concentrations of the integrated stress response inhibitor (ISRIB) that reverses the effect of eIF2α phosphorylation, the PKR inhibitor CAS 608512-97-6 that prevents triggering of the ISR by foreign nucleic acids, the PERK inhibitor CAS1337531-89-1 that halts the ISR triggered by ER stress, and the GCN2 inhibitor CAS 1448693-69-3 that blocks the ISR triggered by tRNA depletion and ribosomal stress for 3 h before harvesting. One representative experiment out of four is shown in the figure. **e** Quantification of the specific bands in the experiment shown in Fig. 4d. The data are shown as the ratio between the intensity of the FLAG-BPLF1 and ATF4 bands after normalization for the intensity of the GAPDH loading control.

To investigate which eIF2α kinase mediates the BPLF1 effect, the upregulation of ATF4 was compared in BPLF1 transfected HeLa cells treated with the integrated stress response inhibitor (ISRIB) that reconstitutes the activity of the translation initiation complex in the presence of phosphorylated eIF2α[59], a PKR inhibitor that prevents

triggering of the ISR by foreign nucleic acids[60], a PERK inhibitor that halts the ISR triggered by ER stress[61], and a GCN2 inhibitor that blocks the ISR triggered by tRNA depletion and ribosome stalling[62]. The accumulation of ATF4 was significantly reduced by treatment with ISRIB, confirming that inactivation of the initiation complex by eIF2α

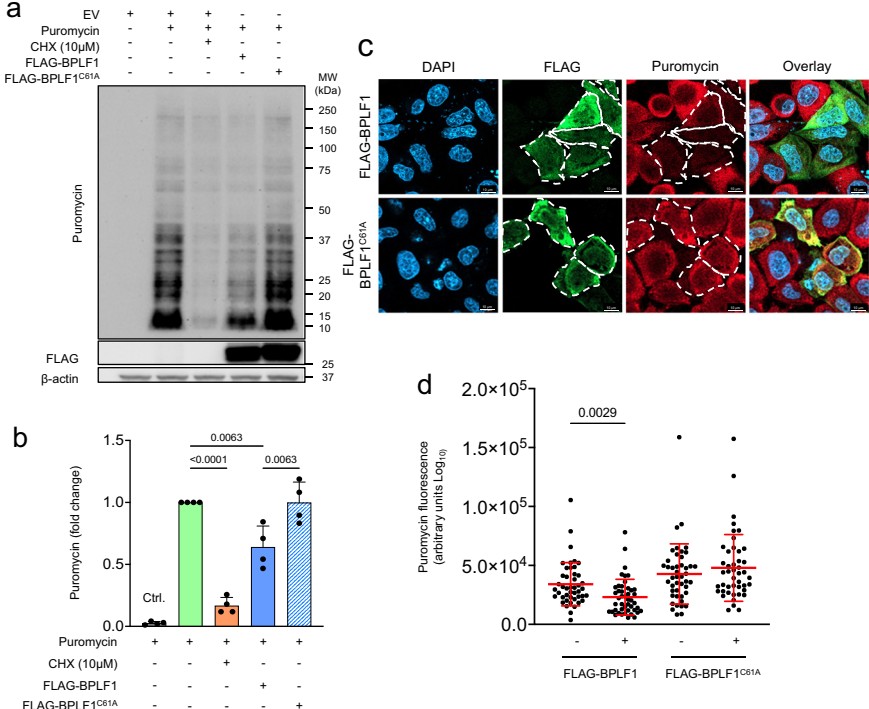

**Fig. 5 | BPLF1 promotes a global inhibition of host protein synthesis. a** HeLa cells transfected with FLAG-EV/BPLF1/BPLF1$^{C61A}$ were cultured for 24 h before analysis of protein synthesis. SUnSET assays were performed by addition of 10 μg/ml Puromycin for 10 min. As a control for protein synthesis inhibition, 10 μM cycloheximide (CHX) was added 5 min before Puromycin treatment. Western blots were probed with a Puromycin-specific antibody. Blots from one representative experiment out of four are shown in the figure. **b** Densitometry quantification of puromycin incorporation. The intensities of puromycin-labeled bands in CHX-treated and BPLF1/BPLF1$^{C61A}$ transfected cells are shown relative to EV untreated control. Mean ± SD of four independent experiments. Statistical analysis was performed using paired Student *t*-test. **c** SUnSET assay was performed in control and FLAG-EV/BPLF1/BPLF1$^{C61A}$ transfected cells grown on cover slides, followed by double staining with FLAG and puromycin-specific antibodies and visualization by confocal microscopy. Representative micrographs illustrating the significant decrease of puromycin fluorescence in BPLF1 expressing cells. Scale bar, 10 μm. **d** Quantification of puromycin fluorescence in BPLF1/BPLF1$^{C61A}$ positive and negative cells from the same transfection experiment. Mean ± SD of fluorescence intensity in 45 individual cells scored for each condition. Significance was calculated by paired, two-tailed, Student *t*-test.

phosphorylation is required for the BPLF1 effect. The upregulation of ATF4 was also inhibited by treatment with the GCN2 inhibitor in a dose-dependent manner. In contrast, the PKR and PERK inhibitors had minor or sometimes enhancing effects poorly reproduced in repeated experiments (Fig. 4d, e). In line with the induction of eIF2α phosphorylation, the expression of BPLF1 was accompanied by a significant decrease in global mRNA translation measured in puromycin incorporation assays (Fig. 5a, b). The effect on translation was further confirmed at the single-cell level by FLAG and puromycin double staining, which revealed weaker puromycin staining in cells expressing BPLF1 compared to BPLF1 negative cells from the same transfection experiments while expression of the BPLF1$^{C61A}$ mutant had no appreciable effect (Fig. 5c, d). These findings support a model where failure to resolve ribosome stalling plays an important role in the inductions of the ISR by the vDUB. The model is substantiated by the finding that reconstitution of cap-dependent translation by treatment with ISRIB did not affect the capacity of BPLF1 to inhibit the ubiquitination of 40 S ribosome protein in ANS treated cells or to stabilize a model RQC substrate (Supplementary Fig. 5a, b). Thus, the inhibition of the RQC in BPLF1-expressing cells appears to be independent of prior ISR activation.

**Endogenously expressed BPLF1 enhances the translation of viral proteins and the release of progeny virus during productive EBV infection**

Reporters containing long poly(A) sequences have been instrumental in elucidating the mechanisms and consequences of ribosome stalling, but protein-coding open reading frames (ORFs) usually do not contain long poly(A) stretches. Naturally occurring mRNA features that may

affect translation elongation include, for example, Guanin-rich domains that may form high-order structures known as G-quadruplex (G4)[63]. The G4-forming sequences generally encode low-complexity amino acid repeats that are frequently found in viral proteins. A prototype example is the ORF of the EBV nuclear antigen-1 (EBNA1) that contains an ~600 nucleotide-long G-rich stretch encoding a Gly-Ala repeat (GAr)[64] (Fig. 6a). The GAr was shown to impair the translation elongation of EBNA1 mRNA both in vitro and in cells, resulting in low protein expression and decreased antigen presentation[65]. Notably, other features of the EBNA1 mRNA may also affect translation efficiency. In particular, during productive infection, EBNA1 is transcribed from the viral Fp promoter, and splicing of the primary transcript joins a short exon located in the BamH1-U fragment of the genome to the first rightward ORF of the BamH1-K fragment[66] (Fig. 6a). The U exon contains an IRES sequence[67] that partially overlaps with an uORF[68], which may promote, alone or in combination, cap-independent translation.

To investigate whether physiological levels of the vDUB may regulate the translation of EBNA1, the productive virus cycle was induced in a pair of lymphoblastoid cell lines (LCLs) established by immortalization of B lymphocytes with recombinant EBV encoding either catalytically active BPLF1 (LCLwt) or the mutant BPLF1$^{C61A}$ (LCLcm)[43]. Since the virus is tightly latent in LCL cells, stable sublines expressing the Dox-regulated EBV transactivator BZLF1 were produced to ensure efficient and synchronous induction[69]. Control and Dox-treated cells were cultured for 72 h before protein and mRNA expression analysis by western blot and qPCR. Comparable induction levels were achieved in the two cell lines, as confirmed by the similar

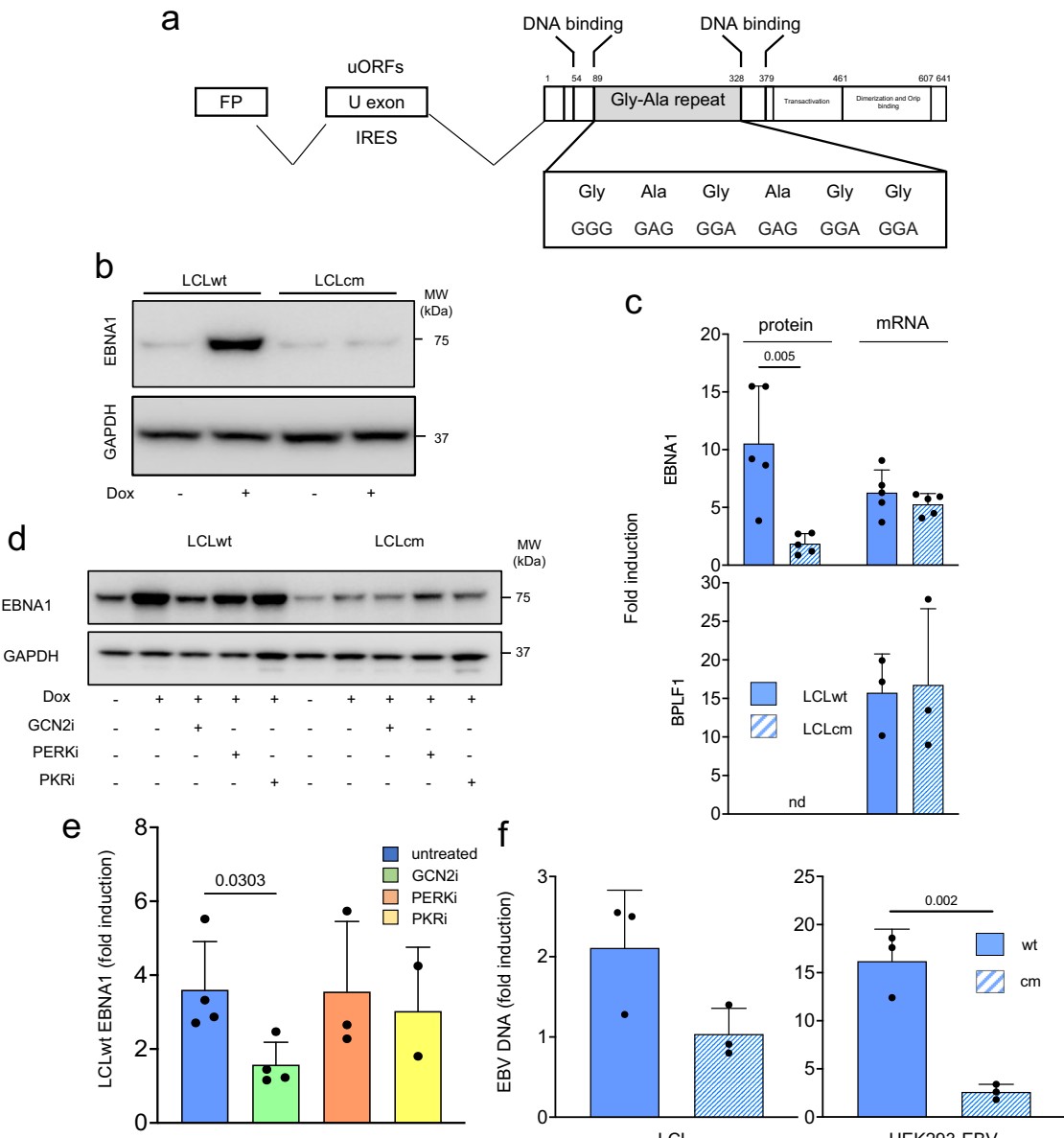

**Fig. 6 | Catalytically active BPLF1 promotes the translation of EBNA1 mRNA and the release of infectious virus particles. a** Schematic illustration of the EBNA1 mRNA transcriptional unit and coding sequence. During productive infection, EBNA is transcribed from the Fp promoter, and exons in BamH-U and BamH-K are joined by splicing. The U exon contains IRES and uORF domains. The amino acid position of the DNA-binding, Gly-Ala repeat, transactivation, dimerization, and Orip binding domains are indicated. The G4 forming GAGAGG motif repeated 13 times in the B95.8-encoded EBNA1 are highlighted. **b** BPLF1 promotes the upregulation of EBNA1. The productive virus cycle was induced in LCLs immortalized with recombinant EBV expressing wild-type (LCLwt) or catalytic mutant (LCLcm) BPLF1. Cell lysates were probed with the indicated antibodies. Blots from one representative experiment out of four are shown. **c** Quantification of the EBNA1 band detected in western blots and qPCR quantification of the corresponding mRNAs. Western blots for BPLF1 were not done due to a lack of suitable antibodies. Fold induction was calculated relative to uninduced cells after normalization to GAPDH for western blots or the *MLN51* housekeeping gene for qPCR. Mean ± SD of four independent experiments. Significance was calculated by paired, two-tailed, Student *t*-test. **d** The upregulation of EBNA1 is dependent on GCN2 activity. The productive cycle was induced in LCLwt/cm in the presence of 1 μM GCN2 inhibitor, 1 μM PERK inhibitor, or 0.3 μM PKR inhibitor. Blots from one representative experiment out of four are shown. **e** Quantification of the intensity of the EBNA1 band in four independent experiments. Mean ± SD fold induction in untreated and inhibitor treated LCLwt. Significance was calculated by paired, two-tailed, Student *t*-test. **f** The amount of released enveloped virus in 3 days culture supernatants from induced LCLwt/cm was quantified by qPCR after treatment with DNase to remove free viral DNA. The mean ± SD fold induction in three independent experiments for each cell pair is shown. Significance was calculated by paired, two-tailed, Student *t*-test.

expression of BZLF1 (Supplementary Fig. 6) and the mRNAs of early and late viral genes (see below). A highly significant increase of EBNA1 protein levels was observed in western blots of cells expressing catalytically active BPLF1, while hardly any change occurred in cells expressing the mutant BPLF1$^{C61A}$ (Fig. 6b). Comparable levels of EBNA1 and BPLF1 mRNA were detected in the two cell lines (Fig. 6c), which,

together with the known resistance of EBNA1 to ubiquitin-dependent proteasomal degradation[70], points to the capacity of BPLF1 to promote the translation of EBNA1 mRNA. This conclusion was further substantiated by the failure to rescue the upregulation of EBNA1 by treating LCLcm cells with the proteasome inhibitor MG132 (Supplementary Fig. 7). Given the effect of BPLF1 on the RQC and ISR

responses, the enhanced expression of EBNA1 may be due to translation elongation across the G4 forming region and/or translation initiation from the uORFs or IRES contained in the U exon. To test whether the latter occurs, GCN2, PERK, and PKR inhibitors were added to the cultures during the induction of the productive virus cycle. The upregulation of EBNA1 was significantly decreased in cells treated with the GCN2 inhibitor, while inhibition of PKR or PERK had only minor effects (Fig. 6d, e). The need for GCN2 activity is consistent with a synergistic effect of BPLF1-mediated inhibition of the RQC and activation of the ISR in regulating EBNA1 protein expression.

To test whether BPLF1 may regulate the expression of other viral proteins, western blots of induced LCLwt/mut were probed with a panel of available antibodies specific for LMP1, a truncated version of which is selectively upregulated during productive infection and is required for efficient virus release[71], the early proteins BMRF1 that encodes the viral DNA processivity factor, BXLF1 encoding the viral thymidine kinase, and BGLF5 encoding the viral alkaline exonuclease, and the late protein BdRF1 encoding the p40 subunit of the viral capsid antigen (VCA). Although the detection efficiency varied considerably due to the quality of the specific antibodies, in all cases, higher protein levels were detected in cells expressing catalytically active BPLF1 despite comparable, or in the case of lytic LMP1, lower levels of the corresponding mRNAs (Supplementary Fig. 8a, b). The expression levels were not affected by treatment with MG132 during the last 6 h before harvesting (Supplementary Fig. 8c, d). A search for putative G4-forming sequences using the QGRS MAPPER server[72] returned one or more putative G4 domains in the coding sequence (Supplementary Table 1). This, together with the frequent occurrence in EBV mRNAs of other translation stalling features, including, for example, nucleotide misincorporations and polyproline or hydrophobic domain coding sequences, suggests that BPLF1 may have a broad effect on the translation of viral proteins. In line with the inefficient expression of proteins required for various steps of the productive cycle, lower amounts of enveloped virus particles were recovered in the culture supernatants of cells expressing the catalytic mutant vDUB (Fig. 6f). It is noteworthy that, while significant, the effect of BPLF1 on virus release was relatively small in LCLs, in line with the tight latency observed in this cell type, but more robust in a previously described pair of HEK293T-EBV cell lines carrying wt and mutant BPLF1[C61A43] where the productive virus cycle was efficiently induced by BZLF1 transfection.

In the final set of experiments, we asked whether the capacity of BPLF1 to regulate the RQC and ISR is shared by the vDUBs encoded by other human herpesviruses. FLAG-tagged versions of the N-terminal vDUB domains of HSV1 UL36 (aa 1-287), HCMV UL48 (aa 1-263), and KSHV ORF64 (aa 1-205) were tested in parallel with FLAG-BPLF1/BPLF1[C61A] for their capacity to stabilize the GFPnonSTOP reporter and upregulate ATF4, as a proxy for impaired RQC and ISR activation, respectively. We have previously shown that the catalytic domain of HSV1 UL36 differs from the homologs encoded by EBV, HCMV, and KSHV in its failure to inhibit the IFN response through the inactivation of the TRIM25 ligase[73] and impair SQSTM/p62-mediated selective autophagy[74]. These effects correlated with significant differences in the amino acid sequence and surface charge of the UL36 vDUB domain, which prevented the interaction with 14-3-3 and presumably a wider range of cellular substrates[75]. In repeated experiments, we observed stabilization of the GFPnonSTOP reporter in cells expressing the vDUB domains of BPLF1, UL48, and ORF64. In contrast, the vDUB of UL36 had no reproducible effect (Fig. 7a, b) despite similar expression levels and comparable enzymatic activity measured by labeling with the Ub-VS functional probe (Supplementary Fig. 9). HSV1 UL36 also failed to promote the upregulation of ATF4 (Fig. 7c, d), pointing to significant differences in the contribution of the vDUBs to the life cycles of human herpes viruses.

## Discussion

A growing body of evidence highlighting the importance of the RQC and ISR in regulating mRNA's translation under stress conditions points to a critical involvement of these surveillance mechanisms in controlling viral infection. Here, we have uncovered an unexpected role of the viral DUBs encoded by EBV and other human herpes viruses in reprograming these cellular functions.

Using a co-immunoprecipitation, mass spectrometry, and validation approach, we have found that the vDUB encoded in the N-terminal domain of the EBV large tegument protein BPLF1 is recruited to protein complexes that regulate mRNA biogenesis and translation and include the RQC ubiquitin ligases ZNF598 and LTN1. It is noteworthy that although BPLF1 is a huge cytosolic protein of more than three thousand amino acids, in infected cells, the small N-terminal catalytic domain is released by caspase cleavage and can diffuse between the nucleus and cytoplasm[42], which underlies its pleiotropic effects on the host cell reprogramming that enables the production of progeny virus. While not necessarily mediated by direct interaction with either ZNF598 or LTN1, the presence of the vDUB in protein complexes containing the RQC ligases points to the regulation of translational stress responses as an important component of the viral host reprogramming strategy. In line with this notion, we found that the vDUB counteracts the ZNF598-mediated ubiquitination RPS10 and RPS20 that is triggered by the sensing of collided ribosomes in cells treated with transcriptional stall-inducing doses of ANS[5,6,8,12,19,53], and rescues aberrant polypeptides generated by the stalling of translation on poly(A) sequences from LTN1- and proteasome-dependent degradation. Inhibition of the RQC may facilitate translation elongation across stall-inducing poly(A) sequences[8,14,23]. Hence, if stalling goes undetected, the ribosome may restart translation after prolonged pausing, albeit with an increased risk of codon frame shifts and amino acid misincorporation[76]. Consistent with the capacity of the vDUB to inhibit the RQC, we found that catalytically active BPLF1 promotes the read-through across the poly(A) sequence of the GFP-VHPK20-RFP reporter, resulting in comparable expression of the VHPK0 and VHPK20 fragments in cells expressing the vDUB, whereas the RFP encoded downstream of the K20 tract was expressed at significantly lower levels. Interestingly, several components of the ER-associated RQC that are triggered by ribosome stalling during the translation of secretory and membrane proteins[77], including members of the signal peptide recognition and receptor complexes, the UFM1 ligase UFL1, and components of the SEC61 translocon and associated OST complex that participate in the ribosome-translocon junction, were also found in the BPLF1 interactome, suggesting a general involvement of the vDUB in the regulation of RQC responses.

The events that link ribosome stalling to its downstream effects are not fully understood. The currently accepted scenario poses that stalling during translation elongation causes a collision with the trailing ribosome, which triggers the coordinated ubiquitination of RPS10 and RPS20 by ZNF598 and initiates ribosome subunit disassembly and the degradation of translation-arrested products[23]. In contrast, stalling at the initiation codon, which does not cause ribosome collision, triggers the ubiquitination of RPS2 and RPS3 by RNF10, which may serve as a triage signal for the turnover of ribosome subunits[20]. The capacity of BPLF1 to counteract the ubiquitination of both RPS10 and RPS20 mimics the activity of cellular DUBs, including USP21 and OTUD3, that serve as physiological regulators of the pathway[20,23]. Conceivably, attenuating the cellular response to difficulties encountered during translation may be beneficial for the translation of viral mRNAs that often contain unusual start sites and features that may hinder elongation, such as nucleotide misincorporation and long repetitive sequences. The stabilization of RQC substrates may also afford an important advantage to the virus because aberrant polypeptides produced during faulty translation are a significant source of the antigenic peptides recognized by the antiviral responses[78].

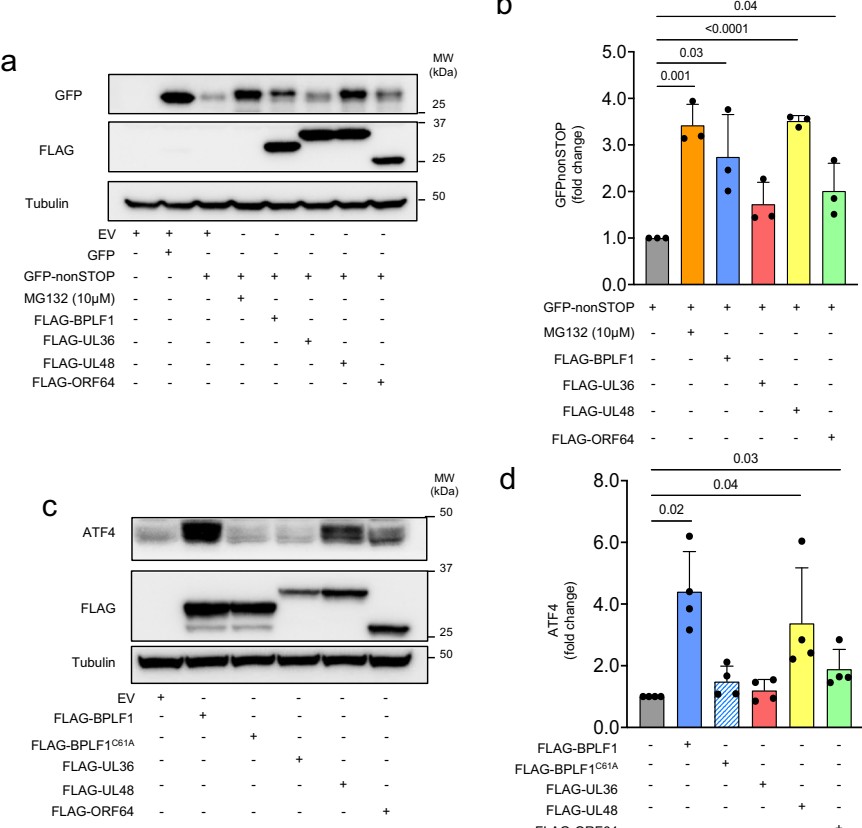

**Fig. 7 | The BPLF1 homologs encoded by human beta and gamma herpesviruses share the capacity to impair the RQC and activate the ISR. a** Western blots of HEK293T cells transfected with a control GFP plasmid, or the GFPnonSTOP reporter together with FLAG-EV, FLAG-BPLF1, or the FLAG-tagged homologs encoded by HSV1 UL36, HCMV UL48, and KSHV ORF64 were probed with antibodies to GFP. As a control for proteasomal degradation, the GFPnonStop transfected cells were treated with 10 µM MG132 during the last 6 h before harvesting. Western blots from one representative experiment out of three are shown in the figure. **b** The intensity of the GFP band was quantified by densitometry in three independent experiments. The mean ± SD relative intensity of the GFPnonSTOP band in transfected cells treated with MG132 or cotransfected with the vDUBs versus control cells is shown. Significance was calculated by paired Student *t*-test. **c** HeLa cells were transfected with FLAG-BPLF1/BPLF1^C61A, the homologs encoded by HSV1, HCMV, and KSHV or the FLAG-empty vector and cultured overnight. The upregulation of ATF4 was assessed in western blots. Blots from one representative experiment out of three are shown in the figure. **d** The intensity of the ATF4-specific bands was quantified by densitometry in three independent experiments. The mean ± SD fold change relative to the untreated control is shown. Significance was calculated by paired, two-tailed, Student *t*-test.

Thus, the effect of BPLF1 may partially explain the paradoxical findings that viral antigens that are abundantly expressed during productive infection are poor targets of EBV-specific cytotoxic T-cells while latent and early antigens are efficiently recognized[79,80]. Notably, ZNF598 was previously shown to be coopted by poxviruses to promote the accumulation of viral proteins[79,80]. Interestingly, while the synthesis of viral proteins was found to be dependent on the expression of ZNF598 and ubiquitination of RPS10 and RPS20, poxvirus infection was still associated with impairment of the RQC, as revealed by the enhanced readthrough of a stall-inducing reporter[79]. The contrasting effects on the activity of ZNF598 point to important differences in the strategies adopted by different viruses for interfering with the RQC.

Impairment of the RQC was shown to trigger the ISR via a direct effect of persistent ribosome stalling on the activation of the eIF2α kinase GCN2 by ZAKα[25,26]. In line with the impaired RQC, we found that BPLF1 promotes the phosphorylation of ZAKα and the ZAKα substrates p38 and JNK that mediate pro-apoptotic ribosomal stress responses. Furthermore, activation of the ISR was confirmed by the accumulation of phosphorylated eIF2α, upregulation of ATF4 and CHOP, and decrease of global protein synthesis in cells expressing the active enzyme. Significantly, the upregulation of ATF4 induced by BPLF1 was inhibited by treatment with ISRIB, which counteracts the effect of eIF2α phosphorylation, and with a specific inhibitor of GCN2,

while minor effects were observed upon inhibition of PERK or PKR. Given the relationship between ribosome stalling, ZAKα activation, and GCN2 phosphorylation, the dependency on GCN2 activation suggests that the induction of prolonged ribosome stalling may play a central role in the translation reprogramming triggered by the vDUB. While the association of viral infection with ISR activation is well known, the role of the ISR in infection is unclear and is expected to vary depending on the virus and type of infection. Like many other viruses, EBV inhibits the ISR triggered by PKR, which prevents a total shutoff of protein synthesis that could severely hinder virus replication[81]. A currently accepted view poses that the preferential shutoff of cellular protein synthesis may be achieved through the degradation of host cell mRNAs by the viral RNase BGLF5[82]. However, the viral enzyme also targets viral mRNAs, and the selectivity mechanism remains unclear. Our findings suggest that upon inhibition of PKR, the activation of a GCN2-dependent ISR may both participate in the shutoff of cellular translation and promote reprogramming of the translation machinery in favor of viral mRNAs that often contain IRES and uORFs. It is noteworthy that uORFs have been identified in all classes of latent, immediate early, early, and late EBV transcripts[68]. Several of these uORFs were shown to downregulate protein expression in luciferase reporter assays, confirming their potential role in regulating EBV translation, but their activity during productive infection was not

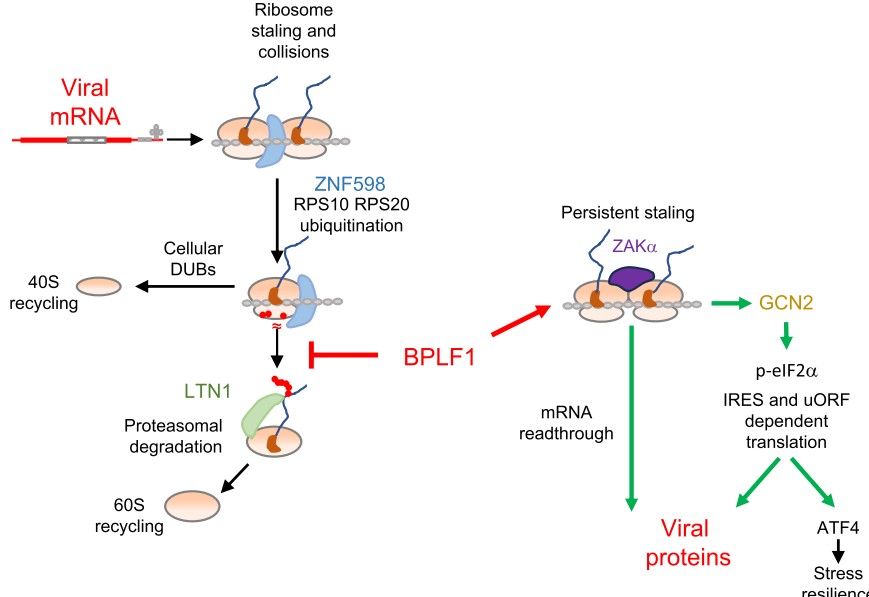

**Fig. 8 | Model of the regulation of viral mRNA translation by the EBV encoded vDUB.** Viral mRNAs often contain structural features that affect the efficiency of translation initiation and elongation and trigger RQC responses. Ribosome stalling and collision occurring during translation elongation trigger the ubiquitination of 40 S ribosome proteins by ZNF598, followed by disassembly of the ribosome, ubiquitination of the 60S-associated polypeptide by LTN1, and degradation by the proteasome, which allow recycling of the ribosome particles. During productive EBV infection, the vDUB encoded in the N-terminal domain of large tegument protein BPLF1 attenuates the RQC response by counteracting the activity of ZNF598, resulting in persistent ribosome staling and the readthrough of stall-inducing mRNAs. Activation of the ZAKα kinase on collided ribosomes triggers the ISR via activation of the GCN2 kinase, which inhibits global protein synthesis and activates uORF and IRES-dependent translation. The readthrough of stall-inducing mRNAs induced by inhibition of the RQC, together with the activation of the ISR, may facilitate the translation of viral mRNAs in productively infected cells.

explored. While uORFs are commonly regarded as translation inhibitors, under stress conditions, uORFs enhance the translation of mRNAs encoding ATF4 and other cellular proteins that regulate the cell cycle and promote stress resilience[32,83]. Thus, activation of the ISR may be instrumental for redirecting the translation machinery towards producing viral proteins and a restricted set of cellular proteins required for cell survival and efficient virus production.

Most relevant for the function of the vDUB during infection is our finding that under physiological conditions of expression, catalytically active BPLF1 enhances the expression of several viral proteins required to produce progeny virus. The highly significant upregulation of EBNA1 is particularly remarkable. EBNA1 is EBV's viral episome maintenance protein and is the only viral protein expressed in all types of infected normal and malignant cells[64]. While the key to EBV persistence during latency, EBNA1 plays an important enhancing role during productive infection[84] when it is expressed from a dedicated lytic promoter that produces transcripts containing IRES[67] and uORF[68] features. In latency, the expression of EBNA1 is maintained at very low levels due to the elongation-hindering effect of a G4 forming sequence in the GAr coding domain[65]. Our findings point to two mechanisms by which BPLF1 could synergistically enhance the translation of EBNA1 during productive infection. By impairing the RQC, BPLF1 may allow a slow-down of translation required to resolve the G4 by the ribosome itself or associated RNA helicases. In addition, the activation of ISR may facilitate translation driven by the IRES and uORFs contained in the mRNA transcribed from the lytic promoter. Interestingly, our analysis of a restricted panel of late viral antigens suggests that the effect of BPLF1 on translation may not be restricted to EBNA1. It is noteworthy that, although a bioinformatics search suggests that G4 forming regions may be relatively frequent in the ORF of EBV mRNAs, other mRNA features, such as nucleotide misincorporation and the presence of long polyproline and hydrophobic domain coding regions, may also constitute translational hinders that may be alleviated by the activity of the vDUB.

Collectively, our findings outline a scenario where, by counteracting the activity of RQC-associated ubiquitin ligases, the vDUB may facilitate the handling of challenging viral mRNAs while decreasing the production of antigenic peptides derived from aberrant translation products. The effect on mRNA translation is likely to be amplified by activation of the GCN2 kinase, which may contribute to the viral-induced shutoff of global protein synthesis while favoring the translation initiation of mRNAs containing features such as uORFs and IRES that are frequently found in viral transcripts (Fig. 8). Interestingly, we found that the capacity of BPLF1 to regulate the RQC and ISR is shared by the homologs encoded by HCMV and KSHV. Genome-wide analysis has revealed an unusually high density of putative G4 forming sequences in the ORFs and regulatory regions of herpesvirus mRNAs[85]. Like EBNA1, the mRNA of the genome maintenance protein of KSHV, LANA1, contains a G4 forming repeat that inhibits translation and antigen presentation[86,87], which underscores the similarity in the lifestyle of these viruses. Interestingly, we found that the vDUB encoded by HSV1 is functionally distinct from the homologs encoded by beta and gamma herpes viruses in that it fails to inhibit the RQC and does not activate the ISR. This, together with our previous findings that HSV1 UL36 fails to regulate the IFN response via interaction with 14-3-3 proteins[75] and does not inhibit p62-mediated selective autophagy[74], pointing to important differences in the tools and strategies evolved by alpha herpesviruses to interfere with ubiquitin regulated processes. The functional divergence of the alpha herpesviruses is further highlighted by the exclusive presence in all members of the family of homologs of the HSV1 ICP0 ubiquitin ligase homologs that play critical roles in virus replication[88]. The functional heterogeneity of the herpesvirus vDUBs is perhaps not surprising given the early branching of the alphaherpesvirus lineage from a common precursor of the beta- and gamma-herpesvirus lineages[89], their different host cell range and substantial differences in the length and regulation of the replicative cycle.

## Methods

### Reagents

For a complete list of reagents, kits, and commercially available or donated plasmids with source identifiers, see Supplementary Data 3. For a list of primers used for cloning and qPCR analysis, see Supplementary Table 2.

### Plasmid construction

To construct the GFPnonSTOP reporter, the stop codon was removed from the GFP-C1 plasmid using PCR primers listed in Supplementary Table 2. The product was religated using the NEbuilder HiFi DNA Assembly Master Mix according to the manufacturer's instructions.

### Tandem mass spectrometry and bioinformatics analysis

The mass spectrometry characterization of the BPLF1 interactome was previously reported[43]. Processed mass spectrometry data and description of the search parameters and acceptance criteria are provided in Supplementary Data 1. For the current analysis, proteins detected by at least four unique spectral counts in biological duplicates of the FLAG-BPLF1 and FLAG-BPLF1$^{C61A}$ immunoprecipitates and either absent or seen at a significantly lower level in duplicate samples of immunoprecipitates from cells transfected with the FLAG-empty vector were considered as positive hits. The bioinformatics resource Search Tool for the Retrieval of Interacting Genes (STRING) v. 9.0[90] and DAVID[91] databases were used to identify the overrepresentation of genes in particular functional categories. Analysis of biological process (BP), molecular function (MF), and cellular component (CC) terms was performed using the ToppCluster[92] and Gene Ontology[93,94] databases. Protein interaction network analysis was performed using the STRING database v. 9.0[90], and the interaction network was visualized using Cytoscape v3.9.1[95].

### Cell lines

A list of the cell lines used in the study in included in Supplementary Data 3. The HeLa and HEK293T lines were cultured in Dulbecco's minimal essential medium (DMEM) supplemented with 10% FBS and 10 µg/ml ciprofloxacin (complete medium) and grown in a 37 °C incubator with 5% $CO_2$. TetOn-BZLF1-EBV immortalized lymphoblastoid cell lines (LCLs) expressing catalytically active or inactive BPLF1[69] were cultured in RPMI1640 medium supplemented with 10% Tet-free FBS. The cells were transfected using either the lipofectamine 3000 or jetOPTIMUS® DNA transfection reagents according to the protocols recommended by the manufacturers. The transfection protocols were optimized for each cell line to achieve at least 50% transfection efficiency. The HEK293T *znf598* knockout cell line was generated by transfecting early passages of HEK293T cells with the plasmid px458_2A_GFP_sgRNA_ZNF598 (gift from Thomas Tuschl) followed by the selection of GFP positive cells by Fluorescence-activated cell sorter (FACS). The knockout efficiency was validated by probing western blots with a ZNF598-specific antibody.

### Induction of the productive virus cycle and quantification of EBV transcripts and released virus

The productive virus cycle was induced by culturing the LCLwt and LCLcm cells for 72 h in the presence of 1.5 µg/ml Doxycycline. Total RNA was extracted with the Quick-RNA MiniPrep kit with in-column DNase treatment according to the recommended protocol. A second DNase treatment was performed to eliminate all traces of EBV DNA, followed by purification with an RNA Clean and Concentrator Kit. One µg total RNA was reverse transcribed using the SuperScript VILO cDNA Synthesis kit and quantified using a LightCycler 1.2 instrument (Roche Diagnostic) with the LC FastStart DNA master SYBR Green I kit and specific primers (Supplementary Table 2). The housekeeping gene *MLN51* was used as an internal control. All reactions were performed in duplicate. The relative fold change of gene expression was determined with the comparative cycle threshold ($2^{-\Delta\Delta CT}$) method. In the HEK293-EBV cell line[69], the productive virus cycle was induced by transfection of a plasmid expressing the EBV BZLF1 transactivator, using Lipofectamine 3000. The release of infectious virus was monitored in culture supernatants cleared of cell debris by centrifugation of 5 min at 20000xg at 4 °C and treated with 20 U/ml DNase I to remove free viral DNA. Viral DNA was isolated using the DNeasy Blood & Tissue Kit (Qiagen, Hilden, Germany), and quantitative PCR was performed with primers specific for a unique sequence in EBNA1 (Supplementary Table 2).

### Immunoblotting and co-immunoprecipitation

For immunoblotting, the cells were incubated for 30 min on ice in NP-40 lysis buffer (50 mM Tris-HCl pH 7.6, 150 mM NaCl, 5 mM $MgCl_2$, 1 mM EDTA, 1% Igepal/rCA-630, 5% glycerol) supplemented with protease inhibitor cocktail, 20 mM NEM and 20 mM iodoacetamide. After centrifugation at 20000xg for 15 min at 4 °C, the protein concentration of the supernatants was measured with a protein assay kit. Equal amounts of lysates were fractionated in acrylamide Bis-Tris 4-12% gradient gel (Life Technologies Corporation, Carlsbad, USA). After transfer to PVDF membranes (Millipore Corporation, Billerica, MA, USA), the blots were blocked in Tris-buffered saline (TBS) containing 0.1% Tween-20 and 5% non-fat milk. The membranes were incubated with the primary antibodies diluted in blocking buffer for 1 h at room temperature or overnight at 4 °C, followed by washing and incubation for 1 h with the appropriate horseradish peroxidase-conjugated secondary antibodies. The immunocomplexes were visualized by enhanced chemiluminescence. For immunoprecipitation, cells were harvested 24 h after transfection and lysed in NP40 lysis buffer supplemented with inhibitors (protease/phosphatase inhibitor cocktail, 20 mM NEM, and 20 mM iodoacetamide) for 30 min on ice. For co-immunoprecipitation, the lysates were incubated with 50 µl of anti-FLAG or anti-Myc conjugated agarose affinity gel for 3 h, at 4 °C with rotation. For co-immunoprecipitation of endogenous ZNF598, the cell lysates were incubated for 4 h with the specific antibodies, followed by the capture of the immunocomplexes with protein-G coupled Sepharose beads. The beads were washed with lysis buffer, and the immunocomplexes were eluted by boiling in 2xNuPAGE Loading buffer supplemented with a sample-reducing agent. All images were acquired using a ChemiDoc Imaging system (Bio-Rad), and the intensity of target bands was quantified using the Biorad ImageLab v. 6.0.1 software.

### Immunofluorescence

The cells were grown on cover slides in 6-well plates for 24 h before transfection. Semiconfluent monolayers were transfected as described and cultured for an additional 24 h. The cells were then fixed in 4% formaldehyde PBS buffer for 20 min, followed by permeabilization for 15 min in 0,1% Triton-X 100, blocking in PBS containing 4% bovine serum albumin (BSA) for 40 min, and incubation with the indicated dilutions of primary antibodies for 1 h at room temperature. After washing 3 × 5 min in PBS supplemented with 0.1% Tween 20 (PBST), the cells were incubated for another 1 h with the appropriate Alexa Fluor-conjugated secondary antibodies. The nuclei were stained with 2.5 µg/ml DAPI in PBS for 10 min, and the cover slides were mounted cell side down on object glasses with Mowiol containing 50 mg/ml 1,4-tdiazabicyclo[2.2.2]octane as an anti-fading agent. Images were acquired using a confocal fluorescence laser scanning Zeiss LSM900 microscope and analyzed using the Fiji ImageJ software.

### Phosphatase treatment

Cells were lysed by incubation for 30 min on ice in NP-40 lysis buffer (50 mM Tris-HCl pH 7.6, 100 mM NaCl, 5 mM $MgCl_2$, 1 mM DTT, 1% Igepal/ CA-630, 5% glycerol) supplemented with EDTA free protease inhibitor cocktail. After centrifugation at 20000xg for 15 min at 4 °C,

the protein concentration of the supernatants was measured with a protein assay kit. The lysates were treated with 1 U/µg protein of bovine intestine alkaline phosphatase for 45 min at 37 °C and the reaction was stopped by adding loading buffer.

## RQC reporter assays

The GFPnonSTOP reporter was co-transfected in HEK293T cells with either FLAG-BPLF1, FLAG-BPLF1$^{C61A}$, or the empty FLAG-vector (EV) as control. After 24 h, the cells were lysed in NP-40 lysis buffer, and protein expression was assessed by western blot. Where indicated, 10 µM MG132 was added to the cultures 6 h before harvesting to inhibit proteasome-dependent degradation.

## Surface sensing of translation, SUnSET assay

HeLa cells were grown in 6-well plates, and semiconfluent monolayers were transfected with FLAG-BPLF1 or FLAG-BPLF1$^{C61A}$ for 48 h. Ten µg/ml of puromycin was added during the last 10 min before harvesting. As a positive control of translation inhibition, the cells were pretreated with 10 µM Cycloheximide (CHX) for 5 min before adding puromycin. Cells were then washed with cold PBS and lysed in NP40 lysis buffer supplemented with a protease inhibitor cocktail. Western blots of lysates from an equal number of cells were probed with a Puromycin-specific antibody. For immunofluorescence, cells grown on cover slides were transfected and treated as described above. The cover slides were then processed for immunofluorescence staining and images were acquired using a fluorescence microscope (Zeiss LSM900). Fluorescence intensity was quantified using the Fiji/ImageJ software.

## Translation-readthrough assay

HEK293T and HEK-ZNF598-KO cells were plated into 6-well plates the day before transfection. Semiconfluent monolayers were co-transfected with the pmGFP-P2A-K0-P2A-RFP (K0) or pmGFP-P2A-K(AAA)20-P2A-RFP (K20) together with FLAG-BPLF1, FLAG-BPLF1$^{C61A}$ or the empty FLAG-vector as transfection control. After culturing for 24 h, one aliquot of the cells was lysed with NP-40 lysis buffer for western blot analysis. GFP and RFP fluorescence was measured in the remaining cells by flow cytometry using a BD LSR II SORP apparatus, and the data were analyzed with the FlowJo software.

## Statistical analysis

Plotting and statistical tests were conducted with data obtained in two or more independent experiments using the Microsoft Excel and GraphPad Prism softwares. No assumptions about data normality were made, and a two-tailed Student $t$-test was used to determine statistical significance.

## Reporting summary

Further information on research design is available in the Nature Portfolio Reporting Summary linked to this article.

# Data availability

All data generated and analyzed during this study are included in this published article and its Supplementary files. Processed mass spectrometry data and details of the databases, softwares, and analysis procedure are provided in Supplementary Data 1. Source data are provided with this paper.

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

## Acknowledgements

We thank Jaap M. Middeldorp, VU University Medical Center, Amsterdam, NL, and Nico Dantuma, Department of Cell and Molecular Biology, Karolinska Institutet, Stockholm, SE, for providing reagents and technical advice. This investigation was supported by grants from the Swedish Cancer Society (contract 211402Pj01H to MGM), the Swedish Research Council (contract 2022-01966 to M.G.M.), and the Karolinska Institutet, Stockholm, SE. F.A.A. was supported by a postdoctoral fellowship awarded by the Wenner-Gren Foundation, Stockholm, SE; F.E. was supported by an Erasmus training fellowship from the University of Coimbra, PT; S.X. was supported by a doctoral fellowship awarded by the China Scholarship Council.

## Author contributions

J.L. and M.G.M. conceived and designed the study; J.L., N.N., C.A.T., F.A.A., and F.E. collected data and prepared figures; S.X. provided analysis tools; J.L. and M.G.M. analyzed the data; M.G.M. supervised the work and provided funding; M.G.M. and J.L. wrote the manuscript with contributions from all authors.

## Funding

## Competing interests

The authors declare no competing interests.
