## [Peer Review File · Nature Communications]

REVIEWER COMMENTS

Reviewer #1 (Remarks to the Author):

In this manuscript, Liu et al. report that the EBV deubiquitinase, BPFL1 associates with ribosomes and counters ubiquitination of RPS10 and RPS3 that are involved in the resolution of stalled ribosomes during ribosome quality control. This they further suggest maximises the synthesis of viral proteins.

Overall, the findings are potentially novel and of broad interest. However, they are currently quite underdeveloped and disconnected, with several bold claims made that are not well supported, apparent inconsistencies in the data and alternative processes that are not considered. These would be important to address, particularly for a journal of this calibre.

Major Points:

The functional role of the EBV DUB in regulating RQC is only very superficially tested. Beyond a general and potentially indirect interaction with ribosomes, the effects on RPS ubiquitination could in fact be caused by effects on eIF2 phosphorylation, not the other way around as supposed. The EBV DUB is a huge protein and has many, many functions associated with its DUB activity including as stated, roles in controlling autophagy and TRIM-mediated IFN responses. It also likely causes cellular stress when expressed on its own outside the context of infection. Any of these could explain the induction of eIF2 phosphorylation which, by slowing initiation rates, could reduce the frequency of ribosome collisions indirectly. There is no experimental evidence of a connection to ZNF598 or RQC as the initiators, rather than them being affected by effects on eIF2 phosphorylation. This concern increases upon seeing the comparison with other herpesvirus DUBs wherein the ones that have no effect are also the ones that don't have these other functions. Indeed, why would the ISRIB/eIF2 phosphorylation inhibitors impact RQC if it triggers, rather than responds to changes in eIF2 phosphorylation? The effects of these inhibitors could simply be by reversing the effect of eIF2 phosphorylation on initiation rates. There are also inconsistencies in several of the results that are hard to reconcile with the overall model, as detailed below.

There are several apparent inconsistencies within the experimental results that would be helpful to clarify as they are confusing. RQC occurs on a small fraction of ribosomes encountering aberrant mRNAs or translation events, as is evident in the small amount of RPS10-Ub and RPS3-Ub compared to the total amount of each protein. If BPFL1 does indeed act on these ribosomes it may explain the small induction of eIF2 phosphorylation in Figure 3A, but how does this result in a relatively robust decrease in overall translation in figure 4A if it is selective? Puromycin/Sunset labelling is not a good way to measure protein synthesis as the puromycin itself blocks translation and directly affects the assay it is supposed to be measuring. This is evident in the fact that very few large proteins can be seen to be labelled and all that is detected are fragments of chains before termination by puromycin. In addition, it seems from figure 4C that only about 20% of cells express the BPFL1 construct, so that would suggest the puromycin-measured effect is very large as it measures the whole culture, most of which are not expressing BPFL1. In figure 5, it seems that BPFL1 fully rescues readthrough of the K20 GFP/RFP reporter by FACS. It would be important to confirm that all of these cells actually express BPFL1 given the

transfection efficiency – I understand this is co-transfection and measurement of the K20 reporter-expressing cells, but co-transfections often result in a relatively large number of cells that only express one but not the other construct and are never quite this efficient. Also, the FACS data doesn't seem to align with the relatively low RFP expression in the Western blots in panel 5D? Related to this, if BPFL1 causes an increase in eIF2 phosphorylation and a large decrease in overall protein synthesis based on puromycin assays, wouldn't this affect GFP expression not just from the K20 reporter but also the KO control? This doesn't seem to be the case but this is hard to judge because the authors only use cells that were not transfected with BPFL1 to set the KO baseline.

Related to this, in many experiments the control condition is omitted for cells expressing BPFL1 or its mutant, and we only see them for the control untransfected line. For example, 2A there is no BPFL1 line that is not treated with ANS, 2C-D there is no BPFL1 line co-transfected with GFP control (they only have GFP-nonstop), 5C-D there is no BPFL1 line with the KO reporter, 7A also does not use a GFP reporter control for the various herpesvirus DUBs. This makes it hard to tell if the vDUBs don't simply affect the baseline of constructs or drug conditions such that the effects may be the same, just starting from a lower baseline in the presence of the DUB.

Figure 1A-B just re-analyse previously published data, but it also highlights the enormous diversity of proteins that BPFL1 apparently interacts with. As a result, the data in panels C-D don't really support later claims that it interacts with ZNF598 in any specific way beyond potentially interacting with ribosomes, along with translocase and many other proteins. While there is a b-actin control in panel d, the GAPDH control is not shown for the IP in panel c. It would be important to show that there are more proteins that BPFL1 does not interact with to demonstrate some level of specificity. But even then, more evidence of an interaction and functional effect on ZNF598 would be needed to support the core claim of the paper and to address the concerns about alternatives driven by eIF2.

Figure 2A-B, Effects on Ub-RPS3 are not particularly convincing and this may make sense as RPS3 ubiquitination is more linked to scanning and initiation. The RPS10-Ub blots are more convincing and would align with a specific effect on ZNF598-mediated processes, although there does seem to be an effect of the BPFL1 mutant; in several cases the blots shown do not mirror the quantification provided particularly well, and maybe a better example could be shown. More broadly, these kinds of analyses are challenging because there is only a very small fraction of the total RPS10 or RPS3 that is modified and even slight misloading or differences in total amounts can make it seem like changes are occurring. The authors should show light, linear exposures of the total for each and normalize to this to get a more accurate measure.

EBNA1 may have G-quadruplexes but it is not clear that the other viral mRNAs do. Why does it affect these other proteins? Could it be due to indirect effects on reactivation as the DUB plays many important roles in virus replication beyond the proposed control of RQC.

Figure 2C-D needs to measure mRNA levels to ensure BPFL1 is not simply stabilizing the mRNA.

Minor:

In the introduction, the authors make several claims about why RQC might function during infection but this jumps back and forth between IRESs and re-initiation, normal host shutoff versus potential roles for ISR. It also mentions that viruses have “challenging” RNAs and deplete tRNAs, but many of the references cited are not virus studies and the general review on the role of tRNAs in viral infection doesn’t mention tRNA depletion resulting in translation issues. It would be helpful to be more specific in discussing the role of ZNF598-mediated RQC which largely operates on elongating ribosomes without mixing it with cap-independent initiation concepts, and provide more direct references supporting statements about viral systems such as tRNA depletion.

Aspects of the discussion should also be adjusted to be clearer and more accurate. There are bold claims made that are not substantiated, such as BPFL1 inactivates ZNF598 or counteracts LTN1-mediated degradation of substrates, which are never experimentally tested. There are recent papers that offer an explanation for the apparent contradiction in RQC activity during poxvirus infection. In general, the discussion is quite open ended and doesn’t present a coherent model. It mentions PKR activation and ISR, but talks about IRESs and alternative initiation etc. But what is the evidence the RQC process being studied here is involved, based on points above, and what’s the overall model for how RQC might control translation of so many different viral mRNAs during EBV infection? Is it through q-quadruplexes in all these mRNAs or broader effects?

The PKR inhibitor blot in 3C seems to suggest there is an effect as good as ISRIB.

BZFL1 should be measured in Figure S1 to ensure the BPFL1 mutant doesn’t simply reduce expression of the reactivator.

In the abstract, perhaps qualify that the tegument protein is BPFL1 in line 3 as the name appears out of nowhere towards the end, with no context.

Reviewer #2 (Remarks to the Author):

Summary:

In this work, the authors show, that viral ubiquitin deconjugase (vDUB), as part of viral tegument BLF1 protein, interferes with events that follow ribosome stalling (and most likely ribosome collision). They first show that vDUB inhibits ubiquitination of 40S ribosomal proteins, an event that triggers ribosome quality control (RQC) and that vDUB promotes translation of otherwise stall-inducing mRNA sequences (like poly A or the "K20" reporter). They also show that translation of other viral proteins that are regulated by stall-inducing mRNA sequences, like for example G-quadruplex, is promoted by vDUB. On top of this they show for a couple of examples that translation of viral proteins is enhanced by vDUB in productively infected cells.

The authors conclude, that hijacking the RQC machinery gives the virus an advantage over the host cell in translating viral mRNA that is in general more prone to stall ribosomes.

The authors also investigate a second cellular event that is triggered by ribosome stalls, that is activation of the integrated stress response (ISR). They convincingly show that vDUB leads to activation of IRS via eIF2 α phosphorylation and decrease of global protein synthesis. Since ISR favors specific translation of mRNA dependent on e.g., specific mRNA secondary structure features, uORFs etc., the authors hypothesize, that this may also favor viral translation. Indeed, they can show that readthrough in a stall-inducing mRNA (translation of EBNA mRNA) coincides with GCN2 activation.

In summary, I find most of the experiments in this paper and the conclusions drawn from them convincing. Moreover, I find this work significant because it describes a so far not well-studied strategy how viruses hijack host translation by balancing out translation quality control pathways to the viruses favor. To me the title "Remodeling of the Ribosomal Quality Control and Integrated Stress Response by Viral Ubiquitin Deconjugases" shows what it promises and I would, in principle, advise for publication. Yet, a few rather minor points should be revised and a few experiments added, if feasible (see detailed comments below).

On top I would like to suggest to address two immediate questions that I personally would find very interesting to be answered, if feasible (not essential for publication). These are:

1.) I wonder, if BLF1 (vDUB) interferes with RQC and ISR directly on the ribosome? I would strongly assume, that it binds to stalled, most likely collided ribosomes and there encounters the players that are initiating downstream quality control processes. Thus, here, I would be very interested so see, if FLAG-tagged BPLF1 co-migrates with ribosomes/polysomes on a sucrose gradient (before and after ANS treatment). RNase I (or S7 nuclease) treatment would even answer the question if BPLF1 binds directly or indirectly to 80S.

2.) One central player that acts at the interface of RQC and ISR on colliding ribosomes (especially when considering prolonged stalls) is ZAK α . While I appreciate, that ZAK α is not found in the MS analysis of the BLF1 interactome, it still doesn't exclude that ZAK α may play a role in the described pathway. Therefore, I would be happy if the authors could address the question, if ZAK α can influence BLF1 (vDUB) activity by any experiment of their choice.

Please find below a few more detailed points/suggestions to address for revision.

Introduction:

With respect to ribosome collision and RQC, the intro seems a little imprecise or out of date. For example:

1.) collided trisomes were analyzed in Matsuo et al., NSMB 2020 (ref 12), not in Ikeuchi et al., (ref 3; this one is for disomes).

2.) for ubiquitination of ribosomal proteins, please add ref Narita et al. Nat Comms 2023 (PMID: 36302773). This one shows that uS10, not eS10 poly-ubiquitination (K63-linked) is required for subsequent splitting by the ASCC. It also shows a human disome structure.

3.) When reporting about the interdependence between ribosome collision, RQC and ISR, the authors should address kinase ZAK α . For example, in Wu et al (cited in ref 21) was shown, that under intermediate doses of anisomycin, eIF2 α phosphorylation is reduced in ZAK KO cells. I consider this quite relevant also for this study, since authors show that vDUB decreases 40S protein ubiquitination exactly after anisomycin treatment.

4.) on page 4 in think RSP2 and RSP3 should be RPS2 and RPS3.

Results:

Related to Figure 1A:

5.) Although not listed, I wonder if ZAK was also a hit in the BPLF1 interactome. Moreover, since UFL1 was found, what about the other two components of the E3 ligase, CDK5RAP3 and DDRGK1? There is some recent work that the UFMylation machinery plays an important role in ER-RQC (e.g., PMID: 3694557). The fact that these proteins are found together with components of protein targeting and translocation to the ER hints at a function of BPLF1 on ribosomes stalled at the ER. Can the authors comment on an enrichment of BPLF1 on ER membranes after anisomycin-mediated ribosome stalling?

6.) eIF5A is not part of the pre-initiation complex (see comment below).

7.) Typo in legend: Western blots (in D)

Related to Figure 2:

8) The Western Blot signals for Ub-RPS3 are rather weak and intensities are hard to compare (could be moved to Supplementary), whereas the signal for Ub-RPS10 is quite clear and effects for BPLF1 are convincing. Yet, I wonder why the authors didn't check of RPS20 (uS10). Especially, according to a recent study (Narita et al. NSMB 2023), uS10 should be the main target of ZNF598 leading to disassembly of collided ribosomes. Could the authors add this experiment?

A comment on weak RPS3 signals: This makes sense in a way, since at least in yeast, RPS3-ubiquitination (by RNF10 homolog Mag2 and Fap1) is rather required to resolve stalled monosomes (e.g. after failed initiation), and less after collision. It is quite likely, that this is similar in humans. Please also check Li et al, Mol. Cell 2022 for discussion (PMID: 36113412).

Related to Figure 3:

9.) I wonder if the increased activation of eIF2 α -phosphorylation by BPLF is in any way correlated with ZAK α activation. Would it be possible to repeat the experiment shown in Fig. 3A and 3C (with GCN2i) in ZAK α KO cells? (Or any other experiment that addresses the question, if ZAK α plays a role for BPLF activity?). I would also be interested in what happens to eIF2 α -P, CHOP and ATF4 expression (Fig. 3A) after ANS addition.

Related to discussion:

10.) On page 15, last paragraph, the authors discuss the series of events following ribosome stalling/collision. I think, that RPS20 ubiquitination is the main event that needs to happen for RQC. Moreover, I think that RPS2/RPS3 ubiquitination are not necessarily following. I'd rather argue that these are independent events and that the nature of the stalled ribosome decides, which r-proteins will be ubiquitinated. The main discrimination here is if a stall induces collisions or not. In case rRNA is defective, initiation will fail and no 80S-80S collisions will occur. This is what -at least in yeast- but likely also in human, leads to RPS3 ubiquitination and later to 18S NRD or autophagy.

I would maybe rephrase chapters in introduction and discussion addressing this (admittedly very complicated) issue.

11.) Would it be possible to add a cartoon or graphic summary of the proposed model for BPLF to support the discussion?

Related to Supplementary Material

12.) Table S1 has many flaws.

- Update functional annotation for UFL1. The main target of the E3 ligase is uL24 of the 60S. uL24-UFMylation is essential for ER-RQC.

- I think that the functional annotation for eIF5A is wrong. It is the EF-P homolog that facilitates translation of poly-Pro stretches. Actually, very recently it was shown to also play a role for RQC and CAT-tailing (PMID: 36804914). This citation needs to be added in intro and should be cited along with ref 44.

- Also the functional annotation of EIF3E2 doesn't match.

- I actually advise to carefully review the entire table.

Reviewer #3 (Remarks to the Author):

Review for the manuscript NCOMMS-23-09344 by Dr. Masucci and co-authors entitled "Remodeling of the Ribosomal Quality Control and Integrated Stress Response by Viral Ubiquitin Deconjugases"

Dr. Masucci and co-authors have reported that the deubiquitinase encoded in the N-terminal domain of the Epstein-Barr virus (EBV) large tegument protein BPLF1 regulates selective autophagy by direct binding to the autophagy receptor SQSTM1/p62. The authors also demonstrated that BPLF1 stabilized SUMOylated topoisomerase-II (TOP2) trapped in cleavage complexes (TOP2ccs), which halted the DNA damage response to TOP2-induced double-strand DNA breaks and promoted cell survival.

In this manuscript, the authors demonstrated that the EBV-encoded BPLF1 regulates RQC and IRS (Integrated stress responses). They demonstrated that RQC factors (collision sensor ZNF598 and 60S associated E3 ligase LTN1) and translation initiation factors (eIF2a, eIF4G1/2), SRP factors (SRP68/Rb) and translocon (Sec63/61B) are co-immunoprecipitated with FLAG-BPLF1 in HEK293T cells (Fig 1). The overproduced virus BPLF1 fragment reduces the levels of ubiquitinated eS10 and uS3, and the downregulation of GFP-polyA reporters was impaired in a deubiquitinating activity-dependent manner (Figs 2 and 5). The overproduced BPLF1 induced ATF4 expression (Fig 3) but inhibited global translation (Fig 4), suggesting the activation of GCN2-mediated IRS. The overproduced BPLF1 facilitates the translation of EBNA1 mRNA that contains G-rich seq during productive EBV infection and the release of infectious virus particles, suggesting DUB inhibits ribosome stalling by G-quadruples sequence. Finally, the BPLF1 homologs of human beta and gamma herpesviruses also impair the RQC and activate the ISR.

The regulation of RQC and IRS by the virus-encoded protein is unique and interesting to an audience of Nature Communications. However, the results in this study are preliminary and indirect evidence for the deubiquitinating activity that regulates RQC activity. More evidence is needed to argue that BPLF1 directly regulates RQC and the IRS.

Major comments.

1. BPLF1 was co-IPed with various factors including RQC factors (collision sensor ZNF598 and 60S associated E3 ligase LTN1) and translation initiation factors (eIF2a, eIF4G1/2), SRP factors (SRP68/Rb) and translocon (Sec63/61B). Therefore, the co-immunoprecipitation of ZNF598 with BPLF1 is not sufficient to demonstrate the direct interaction. The direct binding of BPLF1 to the autophagy receptor SQSTM1/p62 was confirmed by co-immunoprecipitation of transfected BPLF1 and by in vitro affinity isolation of bacterially expressed proteins. BPLF1 also binds and deubiquitinates TOP2. Therefore, it is mandatory to further demonstrate that BPLF1 specifically recognizes ZNF598 using the recombinant proteins and/or the purified protein from the transfected cells.
2. Fig 2A: The role of BPLF1 in the deubiquitylation of RPS10 and RPS3 remains unclear. The overexpression of catalytic-dead BPLF1 also reduced the ubiquitylation of RPS10 and RPS3. More efficient detection of the ubiquitinated RPS20 and RPS3 must be demonstrated to convince the deubiquitylation of ZNF598 targets to regulate RQC. Moreover, the in vitro deubiquitination reaction by BPLF1 using the ubiquitinated RPS20 and RPS3 may clearly demonstrate its activity.
3. There are several possibilities for the reduction of RQC by BPLF1. It may inhibit ZNF598-mediated ubiquitination of the target ribosome proteins in the collided ribosomes. The in vitro ZNF598-mediated ubiquitination assay using the collided ribosome may show the inhibitory function of BPLF1 and BPLF1-C61A. The mutational analysis should be done to show its interaction with ZNF598 is required for BPLF1-mediated RQC function.
4. Fig 2D: The expression levels of GFP-poylA are a well-established indication for ZNF598-mediated splitting of the collided ribosome followed by LTN1-dependent proteasomal degradation of GFP-polyA products. Therefore, the GFP-polyA levels should be compared with the levels in ZNF598-KO and LTN1-KO cells in the same blot.

5. Figs. 3A: To demonstrate the induction of IRS by BPLF1 depends on the inhibition of ZNF598, the BPLF1 overexpression-mediated IRS should be monitored in ZNF598-KO cells. In addition, the quantification of ATF levels in the western blots should be improved by the quantification of the ATF4 levels in the indicated condition using the ATF4-reporter system.

Point-by-point rebuttal

Reviewer #1

The reviewer appreciates the potential novelty and broad interest of the findings but expresses several major and minor concerns:

1. Role of the vDUB on the regulation of the RQC and lack of experimental evidence for the role of RQC inhibition in the triggering of the ISR

Response: We acknowledge that our finding that the vDUB inhibits the ubiquitination of 40S ribosome protein, stabilizes RQC substrates, and activates the ISR did not fully support the model proposed in the manuscript. To address the role of BPLF1 in the ubiquitination of ribosome subunits by ZNF598, we have produced a ZNF598-KO subline of HEK293T that reproduces the phenotype reported in the literature in terms of failure to observe RPS10 and RSP20 ubiquitination in cell treated with low doses of ANS that induced ribosome stalling and collision (**new Figure S1**). As previously reported, the reconstitution of ZNF598 by transfection promotes the ubiquitination of RSP10 and RPS20 in the KO cells without ANS treatment. We now show that this ubiquitination event is inhibited by co-transfection of BPLF1 but not the catalytic mutant BPLF1^{C61A} (**new Figure 2C**).

The ZNF598-KO cell lines also confirmed published data showing that the induction of ribosome collision by ANS treatment promotes the activation of the ZAK kinase in RQC deficient cells (**new Figure S1A, S1B** and Ref # 24, #25). The ZAK kinase was previously shown to connect ribosome stress response with the activation of the IRS via phosphorylation of GCN2 (Ref #24). We now show that expression of the catalytic active BPLF1, but not the mutant, promotes ZAK α phosphorylation (**new Figure 4A**). We believe that these findings, together with the requirement GCN2 for activation of the ISR by BPLF1 (**revised Figure 4D**), provide strong evidence for the capacity of the vDUB to inhibit the RQC, which promotes persistent ribosome stalling and activates the ISR via ZAK α phosphorylation and activation of GCN2.

2. Several apparent inconsistencies are highlighted.

Response: we appreciate that our description of the finding may have been unclear and have extensively modified the text, which we hope will clarify the issues. In particular:

a. The reviewer notices an apparent discrepancy between the weak eIF2 phosphorylation and strong ATF4 stabilization induced by BPLF1.

This is an interesting point for which we envisage two possible explanations: 1. ATF4 is a short-lived protein that is degraded by the proteasome following ubiquitination by CRLs and other ligases; being a potent DUB BPLF1 would stabilize ATF4 once it is efficiently translated following eIF2a phosphorylation magnifying the effect on ATF4 upregulation; 2. The ATF4 transcriptional target GADD34 is, together with PP1, a specific eIF2A phosphatase. Higher phosphatase expression is likely to explain the relatively low levels of phosphorylation upon stabilization of ATF4. We have commented on the issues in the **revised Results page 10, new references 58 and 59**.

b. The reviewer notices an apparent discrepancy between the efficiency of transfection and the effect of BPLF1 on global protein synthesis and questions the suitability of puromycin labeling as a measure of ongoing protein synthesis.

We are a bit puzzled by the comments of the reviewer on the interpretation of the puromycin incorporation assay. In our understanding, the assay assesses translation occurring for 10 min before complete blocking by Cycloheximide and harvesting of the cells. Whether the incorporation of puromycin blocks translation is irrelevant since this is the property used to identify proteins being translated, and the effect will be equal in all samples, independent of BPLF1 expression. The reviewer has a point about transfection efficiency, and we were also a bit concerned about the relatively large effect of BPLF1 in the western blot experiments. This is the reason for including the fluorescence assay shown in **Fig 5C, 5D**, where the intensity of puromycin fluorescence was compared in BPLF1 positive and negative cells from the same transfection experiments. The results confirm that BPLF1 affects global translation. The extent of inhibition in individual cells is difficult to assess since both FLAG-BPLF1 and puromycin-labeled proteins may be expressed at levels below the detection capacity of the immunofluorescence assays. Clearly, BPLF1 does not cause complete inhibition of translation but a shift in translation efficiency that favors certain proteins. Neither the western blot nor the immunofluorescence assays are quantitative, and we are not aiming for precise quantification of the level of protein synthesis but for a qualitative assessment of the presence/absence of inhibition. The description of the data was modified to clarify the issues on **Results page 10**.

c. The reviewer questions the high efficiency of translation readthrough induced by BPLF1 in cells expressing the K20 reporter (old Figure 5, new Figure 3).

We were also amazed by the results of this experiment. We have performed additional experiments, and although less striking, rescue efficiency remains highly significant (**new Figure 3C**). We also show that the rescue level is in the range observed in ZNF598 KO cells (**New Figure S3**). Besides the fact that fluorescence assays based on FLAG immunofluorescence may underestimate transfection efficiency (based on FLAG immunofluorescence, the efficiency of BPLF1 transfection varies in our experience between 30-50%), a likely explanation for the apparent discrepancy between our findings and the levels of rescue reported in the literature is the method used for the analysis of the data. While this type of result is often presented as mean RFP/GFP FL intensity or ratio of RFP/GFP FL intensity, we have used a gating method to identify the cells where read-through occurs. We have chosen to gate because after stalling, the ribosome may resume translation in different frames, which may result in lower RFP fluorescence and underestimate the frequency of readthrough. This rationale is supported by the western blot shown in **Figures 3D, 3E**, where, in BPLF1 expressing cells, the ratio of RFP/GFP band intensity is much lower than the VHP/GFP ratio, which is a marker of readthrough. The reviewer also comments on the lack of controls. In old Fig 5C, the KO value was calculated from the mean % gated cells in Ctr, BPLF1, and BPLF1^{C61A} transfected cells. The column is labeled: KO[ctr, BPLF1, BPLF1^{C61A}] in **new Figure 3C**. We apologize for the mistake. We have added the western blot of KO:Ctr/BPLF1/BPLF1^{C61A} transfected cells (**revised Figure 3D left panel**) and present the quantification data as relative intensity calculated from the blot shown in **5D** after normalization to the level of expression in KO transfected cells (**new Figure 3E**).

3. *The reviewer requests additional controls in several experiments.*

Response: we have added some of the requested controls or explained our argument for not doing so in some cases. In particular:

a. *Figure 1C - the data do not support the claim that the interaction between BPLF1 and ZNF598 is direct.*

We appreciate the reviewer's concern, but we are not claiming that BPLF1 directly interacts with the ligase. To our knowledge, direct interaction with the ligase is not a prerequisite for deubiquitination of the substrate. We have commented on this point on Results page 8. We are confident that new findings on ZNF598-KO cells (new Figure S1) and the ZNF598 reconstitution experiments shown in new Figure 2C establish the capacity of BPLF1 to counteract the ZNF598-dependent ubiquitination of 40S proteins. The requested loading control was included in revised Figure 1C, and LTN1 and ZAK α Co-IPs are now shown in revised Figure 1E.

b. *Figure 2A-B – the effect on 40S ubiquitination is not particularly convincing.*

In this Figure, we analyze the effect of BPLF1 on ribosome ubiquitination induced by ANS treatment. Ubiquitination of RPS10 is not observed upon transfection of BPLF1 alone, as is expected, since the viral enzyme is a DUB and does not have ligase activity. Therefore, we have not included this control. The reviewer also comments on the poor quality of the blots and suggests omitting the data on RPS3 ubiquitination. Following this suggestion, we have performed a new set of experiments showing that BPLF1 inhibits the ubiquitination of RPS10 and RPS20 (new Figures 2A and 2B) and omitted the data on RPS3 ubiquitination.

c. *The reviewer wonders about alternative explanations for the effect of BPLF1 on the translation of EBNA1 and other viral proteins.*

Regarding EBNA1, since the mRNA levels are unaffected, we can safely conclude that the upregulation occurs at the posttranscriptional level. We can also exclude stabilization at the post-translational level since EBNA1 is a very stable protein and, as we and others have repeatedly shown, is not a proteasomal substrate (new Ref #71). Therefore, we are confident in concluding that BPLF1 enhances EBNA1 translation. As for what may influence the translation of other viral proteins, we have at least partially addressed this issue by including new data showing that the viral proteins are not proteasomal substrates (new Figures S5C and S5D) and the ORFs contain one or more putative G4 forming domains (new Table S2). Other features of the viral ORFs, such as the presence of short hydrophobic stretches and poly-proline repeats, may cause ribosome stalling. In this context, an overall effect of BPLF1 in the translation of many EBV mRNAs is not surprising, and a deeper analysis of this issue is certainly interesting and worth doing. However, this is not in the scope of this paper. The reviewer also asks for quantification of BZLF1 expression to ensure that the effects are not due to different activation of the lytic cycle. This control is regularly done. Data are now included in new Figure S3.

d. *The reviewer suggests that the data on the stabilization of RQC substrates shown in Figure 2E should be substantiated by analysis of mRNA expression.*

We appreciate that our data do not formally exclude the possibility that the vDUB may affect mRNA stability. However, we feel that is very unlikely given the lack of effect on the levels of viral mRNAs (see Figures 6C and S5 or the expression of other cellular

proteins (see, for example, ZNF598 in Figure 1C, RPS10, and RPS20 in Figure 2A, and the interactors identified in Figure 1D). The requested experiment was, therefore, not included in the revision.

4. *The reviewer makes several minor comments related to the text and the quality of the blots.*

Response: we have reorganized and extensively modified the text of the Introduction, Results, and Discussion to clarify our arguments. A more illustrative blot is now provided to support the selective effect of the GCN2 inhibitor on the activation of ISR by BPLF1 (**new Figures 4D, 4E**). A new figure documenting the similar expression of the BZLF1 transactivator in Dox-treated LCLwt/cm is now included (**new Supplementary Figure S4**)

Reviewer #2

The reviewer acknowledges the significance of the findings and suggests further experiments addressing the mechanisms by which BPLF1 may activate the RQC and ISR. In particular:

1. *The reviewer wonders whether BPLF1 interferes with the RQC and ISR directly on colliding ribosomes.*

Response: we appreciate that our co-immunoprecipitation experiments do not directly prove that BPLF1 is associated with stalled or collided ribosomes. The sucrose gradient experiments suggested by the reviewer are very interesting, and we plan to set up this methodology to continue our studies on the effect of BPLF1 in translation. However, as the reviewer also points out, the impact of the vDUB on the ubiquitination of RPS10 and RPS20 that is triggered by ribosome stalling (**new Figures 2A, 2B, 2C**), together with our new finding on the interaction of BPLF1 with ZAK α (**new Figure 1D**) and ZAK α phosphorylation in BPLF1 expressing cells (**new Figures 4A**) place the vDUB in the close vicinity if not in direct contact with collided ribosomes.

2. *The reviewer wonders about the role of ZAK in the activation of the ISR by BPLF1.*

Response: we are very grateful for this insightful comment from the reviewer. In response, we have performed additional experiments that we feel have significantly improved the paper and clarified several important issues. We have repeated the co-IPs and found that ZAK α is also present in BPLF1-containing complexes, albeit at low levels, which is in line with the failure to detect the interaction by mass spectrometry (**revised Figure 1D**). Using a ZNF598-KO cell line, we confirmed that ZAK is phosphorylated upon treatment with low doses ANS that promote ribosome collision (**new Figure S1**). We have found that ZAK α is phosphorylated in cells expressing catalytically active BPLF1 (**new Figure 4A**). Together with the effect of the GCN2 inhibitor, these findings support our model where the inhibition of ZNF598-dependent ubiquitination of 40S ribosome proteins by BPLF1 promotes persistent ribosome stalling, which in turn favors the readthrough of stall-inducing mRNAs and activation of the ISR via the ZAK-GCN2 pathway. As suggested by the reviewer, we are illustrating our model in **new Figure 8**.

3. *The reviewer highlights several imprecisions in quoting the references in the Introduction and Discussion.*

Responses: we are grateful for the suggestions. In response, we have extensively revised the Introduction and Discussion of the paper and added the references that were missing in our original manuscript. We have also corrected several printing mistakes in the text and figures.

4. *Specific queries*

Response:

a. *Related to Figure 1 - interaction of BPLF1 with ZAK α and components of the ER-RQC.* We have performed the ZAK co-IP suggested by the reviewer (see above and **revised Figure 1D**). We have also commented on the possible involvement of BPLF1 in the regulation of ER-associated RQC (**Discussion page 19**). Indeed, we have found that BPLF1 interferes with the activity of the UFL1 ligase and regulates the ER-RQC. We have explored this issue in a separate set of experiments that are currently being summarized for a separate publication.

b. *Related to Figure 2 - on the regulation of RPS20 and RPS3 ubiquitination*

As also suggested by Reviewer #1, we have not included the RPS3 ubiquitination data in the revised manuscript but added the data on RPS20 ubiquitination (**new Figure 2A, 2B, and 2C**). The effect of BPLF1 on RPS3 ubiquitination is still an interesting issue that we plan to pursue in a different study.

c. *Related to old Figure 3 - on the role of ZAK α in the activation of the ISR.*

We have performed additional experiments showing that the expression of catalytically active BPLF1 is associated with ZAK α phosphorylation (**new Figure 4A**). Together with the effect of GCN2 inhibitors on the upregulation of ATF4 induced by BPLF1, this finding provides a strong link between inhibition of the RQC with consequent persistent ribosome stalling and the activation of the ISR.

d. *Related to Discussion - a summarizing figure may be added.*

As suggested by the reviewer, we have added a new figure that illustrates our model for the regulation of translation by BPLF1 (**new Figure 8**)

d. *Relate to Table S1.*

The annotation of the table was provided by the Cytoscape software. We have revised and updated the table to the best of our knowledge.

Reviewer #3

The reviewer acknowledges that the regulation of RQC and ISR by a viral DUB is a unique and interesting finding but requests more direct evidence for the involvement of BPLF1 in these effects. In particular:

1. *The reviewer points out that BPLF1 interacts with several components of the RQC pathway, and therefore the co-IP with ZNF598 is no proof of direct interaction.*

Response: we acknowledge that our data do not prove that the interaction of BPLF1 with ZNF598 is direct, and we are not making this claim in the manuscript. However, direct

interaction with the ligase is not a prerequisite for deubiquitination of the substrate by a DUB. We find that: i. BPLF1 is present in protein complexes that mediate the RQC (Figure 1), ii. the active DUB counteracts the ubiquitination of 40S ribosome subunits in cells treated with doses of ANS that promote ribosome stalling (new Figures 2A and 2B) and counteracts the ubiquitination of RPS10 and RPS20 upon reconstitution of ZNF598 in ZNF-KO cells (new Figure 2C), iii. The vDUB stabilizes RQC substrates as observed in ZNF598-KO cells (Figure 2E, new Figure S2). Together, these findings clearly link BPLF1 to the regulation of the RQC. We agree with the reviewer that a more precise mapping of the interaction of BPLF1 with ZNF598, or with any other component that may recruit the vDUB to the complex, is an interesting question that could be approached by *in vitro* pulldown assays using recombinant proteins. However, we believe that this analysis is beyond the scope of the current manuscript, and we have not included the suggested experiments.

2. *The reviewer asks for clarifications on the role of BPLF1 in the deubiquitination of ribosome proteins (Figure 2A).*

Response: We acknowledge that the blots presented in the Figure did not clearly illustrate our conclusions. Following this comment and the suggestions of Reviewers #1 and #2, we have performed a new set of experiments that demonstrate the effect of catalytically active BPLF1 on the ubiquitination of RPS10 and RPS20 in ANS-treated cells (new Figures 2A, 2B). Regarding the efficiency of ANS treatment on RPS10 and RPS20 ubiquitination, the reviewer correctly notices that the effect is weak. This is expected when assaying the ubiquitination of endogenous proteins since treatment with low doses of ANS will induce stalling and collision only in a subset of ribosomes, and the effect will be diminished by the activity of endogenous DUBs. We have clarified this issue on Results page 8. We have substantiated the involvement of the vDUB in the regulation of this ubiquitination event using the ZNF598-KO cell line where reconstitution of ZNF598 expression is associated with ubiquitination of RPS10 and RPS20 in the absence of ANS treatment (new Figure 2C). We have chosen not to perform *in vitro* ubiquitination/deubiquitination assays because, in our experience, this approach will not provide conclusive answers due to the very potent DUB activity of BPLF1 that, taken outside of the cellular context, may deubiquitinate any substrate.

3. *The reviewer requests mutation analysis to corroborate the interaction of BPLF1 with ZNF598.*

Response: As discussed above, we are not claiming that BPLF1 directly interacts with ZNF598. Our data show that the effect on the ubiquitination of 40S proteins, stabilization of RQC substrates, and activation of the ISR correlates with the presence of BPLF1 in ZNF598-containing protein complexes. Identifying the primary interacting partner that recruits BPLF1 to the complexes is an interesting issue that can be approached by the pulldown assays suggested by the reviewer. However, we feel that this information is not essential for interpreting our findings, and we have not included the suggested experiments.

4. *The reviewer requests that the capacity of BPLF1 to stabilize RQC substrates be monitored in ZNF-KO and LTNI-KO cells.*

Response: in Figures 2E, 2F, we show that BPLF1 but not the catalytic mutant rescues the degradation of a bona fide RQC substrate to levels comparable to or even higher than those observed upon inhibition of the proteasome. As now clarified in Results page 9, the parental GFP expressing plasmid was included as a reference for the size of regular. The literature validating GFP-polyA as a bona fide RQC substrate whose degradation is dependent on ZNF598 and LTN1 is extensive. Full validation of the pathways appears to be redundant since both KOs are predicted to result in the stabilization of the substrate, and any additional stabilizing effect of BPLF1 may occur at different stages in the pathway. Therefore, we did not include the requested experiment in the main manuscript but included in the supplementary a blot showing that, as expected, the GFP-polyA reporter is stabilized in ZNF598-KO cells to a level comparable to that achieved by inhibition of the proteasome (new Figure S2).

5. *The reviewer suggests that the effect of BPLF1 on ISR activation be monitored in ZNF598 KO cells.*

Response: we interpret this comment as a request for more direct evidence of the role of RQC impairment in the activation of the ISR response by BPLF1. We believe that this issue is addressed by the new data showing that BPLF1 promotes the autophosphorylation of ZAK α , which activates GCN2 in response to persistent stalling and collision. As shown in new Figure S1, treatment with ANS induces the phosphorylation of ZAK α and activation of the ISR in ZNF598-KO cells, and a similar, albeit weaker, effect on ZAK α phosphorylation is observed in ZNF598 proficient cell upon transfection of the catalytically active vDUB (new Figure 4A). We believe these findings establish a clear correlation between inhibition of the RQC and activation of the ISR and support our model for the effect of BPLF1 illustrated in new Figure 8.

REVIEWER COMMENTS

Reviewer #1 (Remarks to the Author):

In this revised submission, the authors provide additional evidence that BPLF1's DUB activity impacts GCN2 and eIF2 phosphorylation, along with ribosome quality control (RQC) and ribosome read-through on stall reporters. However, some of the new data is not very convincing and more importantly, the authors did not address several key points from the prior reviews. As a result, the findings remain quite preliminary, links to what happens in the true context of infection are limited, and the overall model is largely untested and confusingly speculative.

Specific Points:

Although there is not a lot of new data added, with these revisions the authors convincingly show that there is an effect of BPLF1's DUB activity on eIF2 phosphorylation and induction of a stress response that involves GCN2 and ATF4. Moreover, this appears to contribute to viral protein production. In addition, BPLF1 can counteract the effects of ZF598 expression on RPS10/20 ubiquitination and affects readthrough of stall reporters. However, BPLF1 interacts with an enormous number of different proteins and complexes and has broad effects on the cell, and whether these effects on RQC are direct or an indirect consequence of independent activation of GCN2 remains unclear. As the authors themselves note, the ISR and RQC influence each other's activities, so there may be no direct connections in the reporter assays used in most of the paper. As suggested previously, do things like treatment with ISRIB decrease RQC and reporter readthrough? Is it activation of stress responses and eIF2 that affect RQC, or is it the other way around and RQC is influencing GCN2-eIF2 activation?

The biggest issue with the initial and revised submissions is that they only briefly test the role of BPLF1's DUB activity and GCN2 during infection, but draw sweeping conclusions from limited experiments. Again, it is relatively convincing that GCN2 is important for production of viral proteins, but arguments that DUB activity is regulating viral translation based on the argument the proteins are very stable are not convincing, particularly when studying a ubiquitination regulator with broad functions. Worse still, the authors loosely identify putative GC-rich sequences and build a model that evokes both stalling on these sequences as the driver of translational arrest through RQC and ISR, yet initiation on these same mRNAs occurs using uORFs and IRESs. None of these RNA elements are tested and the model is pure speculation and ultimately, quite confused. Overall, the core of the paper is based on transfection approaches that study a viral protein in isolation and outside the context of infection, but there is very little work done to then relate this to what the authors propose is happening during infection. What makes this even more speculative is the comparisons with other viral DUBs which, as noted before, raises concerns that DUBs that do regulate RQC are doing so largely through indirect effects on stress pathways. Aren't all herpesviruses GC-rich? So, why would only some use this strategy but not others? This data is simply added without any real exploration or context.

Neither of these key points, nor many others, were actually addressed in the revisions. Indeed, the authors used summarization of the prior comments in a way that seemed to write off and ignore many important issues. This includes several important controls to establish baseline effects that also seem applicable to new data added. Overall, rather than boiling reviewer points down to a one-liner that often omits the main point, the authors should have addressed each concern directly in their rebuttal and revisions.

As stated previously, in most experiments the GFP control is only tested in the control condition but it would be important to rule out that BPFL1 doesn't simply increase expression of GFP in general, rather than just the readthrough products. For example, where a control is included, namely the KO control shown in Figure 3D, both GFP and RFP seem increased by BPFL1. This could be hinting at indirect effects of this DUB on protein stability which may dramatically influence reporter activity.

The authors never really address how such large effects are seen with such low transfection efficiency, particularly in co-transfection assays. The images show 2/11 cells expressing, which is hard to reconcile with large effects on overall translation in the culture, and the co-transfection efficiency is unclear, yet FACS show dramatic effects in most expressing cells. I don't understand the gating argument being made for this as it doesn't address co-transfection efficiency.

Why does ANS treatment not induce eIF2 phosphorylation as it should in control samples in Figure S1, despite activation of ZAK as expected? This may suggest that the authors are using the wrong concentration of ANS (see Wu et al, Cell, 2020).

The new data showing ZAK band-shifting in Figure 4A is not convincing as the shifts seem very small, other than in the ANS-treated controls, while BPFL1 and GAPDH all show different band-shifts under different conditions. One could argue the mutant also causes some level of shift, but again there is no untreated control for this sample to set the baseline.

In figure 6B, I understand that there is no antibody available but there is a major assumption that BPLF1 protein is made in the reactivation systems. Since the authors went to the trouble of making a viral mutant it seems odd that they wouldn't include a Flag tag to confirm protein production, and at least similar levels between wildtype and mutant forms. I also could not find any details of how the mutant or cell lines were made, or any reference to their construction to refer to.

In this reviewer's opinion, this manuscript remains potentially interesting but still very underdeveloped. It currently consists of a series of observations and parallels that are loosely stitched together without

any real functional testing of the final model, which is ultimately very speculative. It is clear GCN2 is important, but any role for virus-induced RQC and how this happens remains unclear.

Reviewer #2 (Remarks to the Author):

This is the revised version of the paper “Remodeling of the Ribosomal Quality Control and Integrated Stress Response by Viral Ubiquitin Deconjugases” by authors Jiangnan Liu, Noemi Nagy, Carlos Ayala-Torres Francisco Aguilar-Alonso, Francisco Esteves, Shanshan Xu and Maria G. Masucci.

Compared to the first submission, the revised manuscript is significantly improved, reads much better and the story appears far more conclusive. The authors addressed most of my points and performed several important experiments that strengthen the paper. The most significant improvement is the investigation of the role of collision sensor ZAK-alpha. The authors provide co-IPs demonstrating that ZAK-alpha is indeed present in BPLF1-containing complexes and they show that ZAK-alpha is phosphorylated upon ribosome collision. Of course, one of the immediate follow-up questions would be, if in their system, ZAK-alpha phosphorylation also leads to activation of RSR (ribotoxic stress response), that leads to p38 and JNK-Phosphorylation. Do the authors have any data on that?

Another interesting question for me that was not clarified is, if BPLF1 is indeed associated with collided ribosomes. This could be easily tested by using sucrose density gradient centrifugation. Since the authors are only planning to set the methodology up, I assume this will be beyond the scope of this paper to include the results of these experiments?

Independent of if such experiments still can be included, the results presented here are leading to an interesting variation of how viruses hijack the translation machinery, namely by exploiting the responses of the system to ribosome collisions. By counteracting the immediate response on prolonged collision, RQC, and triggering responses to more persistent stalls, ISR (and RSR?) the virus gain an advantage for translation of its own mRNA over host cellular mRNA. I would thus fully support publication in Nature Communications, if a few minor comments will be addressed:

Minor comments:

- 1.) In case no experimental data on RSR are available, please mention this pathway in introduction and discussion, because it seems very likely, that it is activated after ZAK-alpha phosphorylation.

2.) Line 445: Please check wording. Should it be “contradicting”? Or “contrasting”?

3.) Fig. S2: Legend should be “The GFPnonSTOP reporter is stabilized in ZNF598-KO cells”

4.) Table S1: Please re-check for correctness. For example shouldn't KHSRP be “KH-type splicing regulatory protein”?

Reviewer #3 (Remarks to the Author):

Review for the manuscript NCOMMS-23-09344A by Dr. Masucci and co-authors entitled "Remodeling of the Ribosomal Quality Control and Integrated Stress Response by Viral Ubiquitin Deconjugases"

The authors have addressed most of my previous concerns. I support the publication of manuscript in Nature Communications.

Point-by-point rebuttal

Reviewer #1

We are very sorry that the reviewer perceived our attempt to conceptualize our response to the critique provided to the first version of the manuscript as careless. We sincerely apologize for giving this impression. To avoid further misunderstanding, we will respond in detail to the first and second sets of comments.

1st review major comments.

1. *“The functional role of the EBV DUB in regulating RQC is only very superficially tested. Beyond a general and potentially indirect interaction with ribosomes, the effects on RPS ubiquitination could, in fact be caused by effects on eIF2 phosphorylation, not the other way around as supposed. The EBV DUB is a huge protein and has many, many functions associated with its DUB activity including as stated, roles in controlling autophagy and TRIM-mediated IFN responses. It also likely causes cellular stress when expressed on its own outside the context of infection. Any of these could explain the induction of eIF2 phosphorylation which, by slowing initiation rates, could reduce the frequency of ribosome collisions indirectly. There is no experimental evidence of a connection to ZNF598 or RQC as the initiators, rather than them being affected by effects on eIF2 phosphorylation. This concern increases upon seeing the comparison with other herpesvirus DUBs wherein the ones that have no effect are also the ones that don’t have these other functions. Indeed, why would the ISRIB/eIF2 phosphorylation inhibitors impact RQC if it triggers, rather than responds to changes in eIF2 phosphorylation? The effects of these inhibitors could simply be by reversing the effect of eIF2 phosphorylation on initiation rates. There are also inconsistencies in several of the results that are hard to reconcile with the overall model, as detailed below.”*

a) **Rev. The EBV DUB is a huge protein and has many, many functions associated with its DUB activity.**

Resp: Indeed, BPLF1 is a huge protein with many functions during productive infection. However, we have previously shown that in productively infected cells, BPLF1 is processed by caspases, which releases a functional N-terminal catalytic domain (see Ref#42). Based on this finding, we are confident in the relevance of our functional analysis of the isolated catalytic domain that can diffuse between the nucleus and cytoplasm. The reviewer also points out that we and others have previously shown that the vDUB affects many cellular functions. This is not surprising, given the extensive cellular remodeling associated with productive infection. In normal cells, approximately 200 DUBs control functions regulated by ubiquitin and ubiquitin-like polypeptides. Because of the limited coding capacity, it seems reasonable that the virus would use a single multifunctional DUB to subvert a relevant subset of cellular processes.

b) **Rev: There is no experimental evidence of a connection of ZNF598 or RQC as initiator, rather than the being affected by effects on iEF2a phosphorylation.**

Resp: We have addressed the issue of the direct involvement of BPLF1 in the decrease of 40S protein ubiquitination and the effect on ISR activation by performing two sets of experiments:

- i. by producing a ZNF598-KO cell line we demonstrate a direct impact of the vDUB on the ubiquitination of RPS10 and RPS20 (new Figures S2 and 2C). We believe that this

finding conclusively indicates that the vDUB can inhibit the RQC response triggered by translational hurdles independently of ISR activation.

ii. by showing that expression of the catalytically active vDUB promotes the autophosphorylation of ZAK α that is activated by the interaction with collided ribosomes, which triggers the ISR by phosphorylating eIF2 α (new Figure 4A). This finding, together with the demonstration that only a GCN2 inhibitor prevents the activation of ATF4 in cells expressing the catalytically active vDUB, supports our model of ribosome collision as the primary trigger of ISR. It is noteworthy that other potential GCN2 activators, including, for example, amino acid starvation, ultraviolet light, or proteasome inhibition, are unlikely to be relevant in our experimental setup. Furthermore, the experiments shown in Figures 4D and 4E exclude the involvement of the other known eIF2 α kinases, such as PKR and PERK, that could mediate BPLF1-induced stress responses triggered by inference with other cellular stressors and cellular signaling pathways.

- c) **Rev: This concern increases upon seeing the comparison with other herpesvirus DUBs wherein the ones that have no effect are also the ones that don't have these other functions.**

Resp: The reviewer correctly notices that HSV1 UL36 differs from the homologs encoded by KSHV, HCMV, and EBV in that it does not only fail to regulate the RQC and ISR, as shown in the manuscript, but also regulates the IFN response via inactivation of TRIM25 (see Ref #43) and autophagy (see Ref # 74). These are interesting findings. We have previously shown that the failure to inactivate TRIM25 is explained by a very different surface charge of the HSV vDUB compared to the homologs (see Ref #73), which impairs the interaction with 14-3-3 and presumably many other cellular proteins. The finding that herpesvirus vDUBs have different substrates and functions is not surprising given the diverse host range and type of infection (a very rapid replicative cycle for HSV, compared to a slower or slower cycle for the homologs). That alpha-herpesviruses adopt unique strategies for interfering with ubiquitin/UbL-regulated processes is also illustrated by the fact that only this virus family encodes a viral ubiquitin ligase (HSV ICPO and homologs). While a deeper analysis of the mechanism by which HSV1 may regulate the RQC and identification of the unique substrates of UL36 are undoubtedly interesting, we firmly believe that the different behavior of the HSV1 vDUB compared to the homologs does not challenge the significance of our findings. We have commented on this issue in the revised Discussion (Discussion page 24)

- d) **Rev: Indeed, why would the ISRIB/eIF2 phosphorylation inhibitors impact RQC if they trigger, rather than respond to changes in eIF2 phosphorylation?**

Resp: our findings do not imply that impairment of the ISR by ISRIB, or the GCN2 inhibitors should affect the regulation of the RQC by BPLF1, and we have not made this suggestion. While our data and the available literature support a scenario where inhibition of the RQC activates the ISR via induction of persistent ribosome stalling, autophosphorylation of ZAK α , and activation of GCN2, it seems very unlikely that the reactivation of cap-dependent translation induced by ISRIB or GCN2 inhibitors would affect the capacity of the vDUB to counteract the ubiquitination of ribosomal proteins. The reviewer correctly points out that activation of the ISR could, by inhibiting global translation, decrease the RQC load and reduce the steady-state levels of 40S ribosome ubiquitination. However, it is unlikely that this would prevent ribosome stalling in ANS-treated cells or translation elongation across stall-inducing sequences by itself.

2. *"There are several apparent inconsistencies within the experimental results that would be helpful to clarify as they are confusing. RQC occurs on a small fraction of ribosomes encountering aberrant mRNAs or translation events, as is evident in the small amount of RPS10-Ub and RPS3-Ub compared to the total amount of each protein. If BPFL1 does indeed act on these ribosomes it may explain the small induction of eIF2 phosphorylation in Figure 3A, but how does this result in a relatively robust decrease in overall translation in Figure 4A if it is selective? Puromycin/Sunset labeling is not a good way to measure protein synthesis as the puromycin itself blocks translation and directly affects the assay it is supposed to be measuring. This is evident in the fact that very few large proteins can be seen to be labeled and all that is detected are fragments of chains before termination by puromycin. In addition, it seems from Figure 4C that only about 20% of cells express the BPFL1 construct, so that would suggest the puromycin-measured effect is very large as it measures the whole culture, most of which are not expressing BPFL1. In Figure 5, it seems that BPFL1 fully rescues readthrough of the K20 GFP/RFP reporter by FACS. It would be important to confirm that all of these cells actually express BPFL1 given the transfection efficiency – I understand this is co-transfection and measurement of the K20 reporter-expressing cells, but co-transfections often result in a relatively large number of cells that only express one but not the other construct and are never quite this efficient. Also, the FACS data doesn't seem to align with the relatively low RFP expression in the Western blots in panel 5D? Related to this, if BPFL1 causes an increase in eIF2 phosphorylation and a large decrease in overall protein synthesis based on puromycin assays, wouldn't this affect GFP expression not just from the K20 reporter but also the KO control? This doesn't seem to be the case but this is hard to judge because the authors only use cells that were not transfected with BPFL1 to set the KO baseline."*

- a) **Rev: If BPFL1 does indeed act on these ribosomes it may explain the small induction of eIF2 phosphorylation in Figure 3A, but how does this result in a relatively robust decrease in overall translation in figure 4A if it is selective?**

Resp: We appreciate the concern of the reviewer. However, the relatively low levels of eIF2a phosphorylation observed at steady-state in cells expressing the vDUB are likely explained by a BPLF1-mediated enhancement of a feedback loop that, via stabilization of ATF4, promotes the activation of the eIF2a phosphatase (see comments on Results page 12). Our puromycin incorporation assay indicates an average 40% reduction of overall translation, which, despite variations between experiments, is significant but lower than the 90% reduction observed in CHX-treated cells (see Figure 5B). Notably, puromycin labeling is performed in a 10-minute pulse and assesses the abundance of proteins that are being synthesized when translation is stopped by puromycin incorporation. At the steady state, the effect of the vDUB on the expression of individual proteins is likely to vary depending on mRNA abundance, rates of translation, and turnover.

- b) **Rev: it seems from Figure 4C that only about 20% of cells express the BPFL1 construct, so that would suggest the puromycin-measured effect is very large as it measures the whole culture, most of which are not expressing BPFL1**

Resp: As indicated above, the average inhibition of protein synthesis measured in repeated experiments is approximately 40%. This level of reduction is well in line with that observed in fluorescence assays where the red (puromycin) and green (BPLF1) fluorescence were quantified in individual cells (see Figure 5D). Based on the

immunofluorescence shown in Figure 5C, the reviewer estimates a transfection efficiency of 20%, which does not account for the magnitude of the effect on translation. The figure was mounted from a selected field containing both transfected and non-transfected cells to illustrate the impact of BPLF1 on puromycin incorporation. We acknowledge that the figure may suggest low transfection efficiency, but this is not true. We have calibrated our transfection procedures for different cell lines to regularly achieve at least 50% transfection efficiency. To avoid further misunderstanding, this is specified in the new revision, and more presentative images are included (see new Figure 5C upper panels)

- c) **Rev: In Figure 5, it seems that BPFL1 fully rescues the readthrough of the K20 GFP/RFP reporter by FACS. It would be important to confirm that all of these cells actually express BPFL1 given the transfection efficiency.**

Resp: We appreciate that the results shown in the Figure indicate a high level of co-transfection since the active vDUB promotes very efficient readthrough as assessed by the rescue of RFP fluorescence and VHP/GFP ratio. This level of rescue was reproduced in repeated experiments and is only observed in cells expressing the active vDUB despite comparable, or in cases superior, expression of the catalytic mutant assessed in western blots (see, for example Figures 2A, 2E, 4A). Assessment of transfection efficiency by BPLF1 immunofluorescence is undoubtedly feasible but will not change the findings. Notably, the similar levels of RFP fluorescence in the K20 control and BPLF1^{C61A} transfected cells exclude that the rescue observed in cells expressing the active vDUB may be an artifact of inappropriate gating and leakage of the GFP fluorescence in the RFP channel. We comment in the next section on the high efficiency of RFP rescue measured by FACS and assessed by the gating method. Notably, although the percentage of cells falling in the readthrough quadrant increases from a mean of 20% in control and BLF1^{C61A} transfected cells to a mean of 70% in BPLF1 transfected cells, corresponding to an average 3.5-fold increase, the peak of RFP fluorescence remains approximately 10-fold lower in the K20-BPLF1 transfected compared to KO-[EV, BPLF1, BPLF1^{C61A}] transfected cells (see Figure 3A). The reviewer comments that, in co-transfection experiments, the two plasmids may not be co-expressed in all cells. This is undoubtedly true, but only GFP-positive cells were included in the FACS analysis, which decreases the proportion of single transfectants. Furthermore, only catalytically active BPLF1 rescues RFP fluorescence in spite of comparable expression of the active and inactive vDUB as assessed by western blot analysis of the same experiment (see Figure 3C)

- d) **Rev: the FACS data doesn't seem to align with the relatively low RFP expression in the Western blots in panel 5D?**

Resp: As discussed in the manuscript, the low levels of RFP are most likely explained by the random restarting of translation in different frames, which would result in the expression of truncated non-fluorescent species and an overall decrease of RFP fluorescence also in cells where readthrough occurs (see Results pages 11-12). This phenomenon has been amply documented upon inhibition of the RQC (see for example Ref 55). The difference between the efficiency of rescue assessed by the gating of GFP-RFP fluorescence and the western blot analysis is explained by the higher sensitivity of the fluorescence-based assay and the gating method used to assess rescue, which scores as rescue also a relatively small increase of RFP fluorescence.

e) Rev: if BPFL1 causes an increase in eIF2 phosphorylation and a large decrease in overall protein synthesis based on puromycin assays, wouldn't this affect GFP expression not just from the K20 reporter but also the KO control?

Resp: we have included the western blots of the KO transfected EV, BPLF1, and BPLF1^{C61A} transfected cells (see Figure 3D right panels). The blots clearly illustrate the similar expression of GFP in BPLF1 transfected cells compared to vector and BPLF1^{C61A} transfected cells. This is consistent with the results of the puromycin assays since, as discussed above, the ongoing translation detected by the puromycin pulse assay will impact the steady-state levels of protein differently depending on their rate of synthesis and turnover. Importantly, comparable levels of GFP were observed in all K20 transfectants (Figure 3D), which excludes a systematic effect of BPLF1 on the steady-state expression of the reporter. To further exclude possible biases, in the revised Figure 3E, the levels of GFP expression in KO transfected cells were used to normalize the calculation of the GFP/RFP and GFP VHP ratios.

3. " Related to this, in many experiments, the control condition is omitted for cells expressing BPFL1 or its mutant, and we only see them for the control untransfected line. For example, in 2A there is no BPFL1 line that is not treated with ANS, 2C-D there is no BPFL1 line co-transfected with GFP control (they only have GFP-nonstop), 5C-D there is no BPFL1 line with the KO reporter, 7A also does not use a GFP reporter control for the various herpesvirus DUBs. This makes it hard to tell if the vDUBs don't simply affect the baseline of constructs or drug conditions such that the effects may be the same, just starting from a lower baseline in the presence of the DUB."

a) Rev: control condition is omitted for cells expressing BPFL1 or its mutant, and we only see them for the control untransfected line.

Resp: We thank the reviewer for pointing out this blunder. In fact, in all western blots images, the (-) refers to the lack of BPLF1/BPLF1^{C61A} expression, but, as stated in Material and Methods, Results and Figure Legends, the control cells were in all cases transfected with the empty vector. We apologize for the lack of clarity. This was now corrected in all figures in manuscript.

b) Rev: For example, 2A there is no BPFL1 line that is not treated with ANS

Resp: The figure aims to document the capacity of BPLF1 to inhibit the ubiquitination of ribosome proteins by ZNF598. Since BPLF1 is a DUB, we do not expect to see enhanced ubiquitination of ribosome proteins in BPLF1 transfected cells. In the revised manuscript, we have documented the deconjugase activity of BPLF1 by showing that BPLF1 deubiquitinates RPS10 and RPS20 in ANS-treated cells and ZNF598-KO cells when ubiquitination is induced by overexpression of the ligase (see Figures 2A, 2B and 2C). Since the control will not be informative, it was not included in the revision.

c) Rev: 2C-D there is no BPFL1 line co-transfected with GFP control (they only have GFP-nonstop)

Transfection with the parental GFP plasmid was included in the analysis to provide a size reference for regular GFP, which is required to determine whether the stabilized GFP-non-STOP product is derived from elongation into the poly(A). The figure shows that a larger GFP species accumulates in cells treated with MG132, expressing the catalytically active but not the mutant vDUB (see Figure 2E). As discussed above, BPLF1 has no consistent effects on the expression of GFP after transfection for 24h or 48h.

Our data also indicate that BPLF1 does not have a general enhancing effect on transcription (see Figures 6C and S5C). The impact of BPLF1 on the expression of parental GFP is irrelevant to our conclusion on the stabilization of the RQC reporter. Therefore, we have not included this control.

d) Rev: 5C-D there is no BPFL1 line with the KO reporter

Resp: This control was included in the revised manuscript together with a new quantification of the data where the level of expression of the GFP in the K20 transfectant was normalized to the level of expression in the KO transfectants, which corrects for possible non-specific differences in the expression of the reporters (see Figures 3D, 3E)

e) 7A also does not use a GFP reporter control for the various herpesvirus DUBs.

See comment to point c). GFP is used in this blot as a size reference for regular GFP. The figure shows that except for HSV UL36, the active vDUBs stabilize a product of slightly larger size compared to regular GFP, which corresponds to elongation through the poly(A) tail (see Figure 7A).

4. *"Figure 1A-B just re-analyze previously published data, but it also highlights the enormous diversity of proteins that BPFL1 apparently interacts with. As a result, the data in panels C-D don't really support later claims that it interacts with ZNF598 in any specific way beyond potentially interacting with ribosomes, along with translocase and many other proteins. While there is a b-actin control in panel d, the GAPDH control is not shown for the IP in panel c. It would be important to show that there are more proteins that BPFL1 does not interact with to demonstrate some level of specificity. But even then, more evidence of an interaction and functional effect on ZNF598 would be needed to support the core claim of the paper and to address the concerns about alternatives driven by eIF2."*

a) Rev: the data in panel C-D don't really support later claims that it interacts with ZNF598 in any specific way beyond potentially interacting with ribosomes, along with translocase and many other proteins.

Resp: We are not claiming that BPLF1 interacts directly with ZNF598, but the vDUB is present in protein complexes that contain ZNF598 and counteracts the activity of the ligase. To our knowledge, direct interaction with the ligase is not a prerequisite for deubiquitination of the substrate. The new data obtained with a ZNF598-KO cell line confirm the capacity of BPLF1 to directly counteract the activity of the ligase (see Figure 2B). While interesting, in vitro pulldown assays aiming at identifying the primary BPLF1 interactor that recruits the vDUB to the complex are outside of the scope of this manuscript.

b) Rev: While there is a b-actin control in panel d, the GAPDH control is not shown for the IP in panel c.

Resp: The controls were included in the revised figure (see Figure 1C)

c) Rev: It would be important to show that there are more proteins that BPFL1 does not interact with to demonstrate some level of specificity.

Resp: Our validation experiments aimed to confirm the interaction with candidates identified in repeated immunoprecipitation and mass spectrometry analysis. Some additional proteins were included based on the functional networks identified by the mass spectrometry data. Therefore, it is not surprising that all candidates are found to be interactors. Lack of interaction is generally difficult to demonstrate in the

absence of candidates. Still, we can say that BPLF1 does not interact with any protein since β -actin and GAPDH are not found in the immunoprecipitates (see Figures 1C, and 1D).

- d) **Rev: more evidence of an interaction and functional effect on ZNF598 would be needed to support the core claim of the paper and to address the concerns about alternatives driven by eIF2.**

Resp: We have addressed the functional interaction of BPLF1 with ZNF598 by confirming the capacity of BPLF1 to counteract the ZNF598-dependent ubiquitination of RPS10 and RPS20 in transfected ZNF598-KO cells (see Figure 2C). Furthermore, we have substantiated our model suggesting that the persistent ribosome collision induced by inhibition of the RQC triggers the ISR by showing that BPLF1 promotes the autophosphorylation of ZAK α that was shown to occur upon dimerization of the kinase on collided ribosomes (see Figure 4A). This, together with the demonstration that the ISR triggered by BPLF1 is dependent on phosphorylation of eIF2 α by the GCN2 kinase that is activated by ZAK α (see Figures 4D and 4E), identifies the “ribosome collision-pZAK α -pGCN2-peIF2 α ” pathway as the signaling pathway involved in the activation of the ISR. Our data also exclude the involvement of the other two kinases known to phosphorylate eIF2 α : PKR (triggered by viral infections) and PERK (triggered by ER stress responses). The reviewer suggests that eIF2 α phosphorylation may be the primary driver of the observed phenotype, but in the absence of a candidate kinase, we don't see how we could prove/disprove the alternative scenario.

5. *“Figure 2A-B, Effects on Ub-RPS3 are not particularly convincing, and this may make sense as RPS3 ubiquitination is more linked to scanning and initiation. The RPS10-Ub blots are more convincing and would align with a specific effect on ZNF598-mediated processes, although there does seem to be an effect of the BPLF1 mutant; in several cases, the blots shown do not mirror the quantification provided particularly well, and maybe a better example could be shown. More broadly, these kinds of analyses are challenging because there is only a very small fraction of the total RPS10 or RPS3 that is modified and even slight misloading or differences in total amounts can make it seem like changes are occurring. The authors should show light, linear exposures of the total for each and normalize to this to get a more accurate measure.”*

Response: we have addressed these concerns by performing a new set of experiments that confirm the capacity of BPLF1 to inhibit the ubiquitination of RPS10 and RPS20 in ANS-treated cells and in ZNF598-KO cells that overexpress the ligase (see Figure 2A, 2B, 2C). Light exposures of the of the RPS10 and RPS20 blots are also included. As suggested by the reviewer, we have omitted the data on RPS3.

6. *“EBNA1 may have G-quadruplexes but it is not clear that the other viral mRNAs do. Why does it affect these other proteins? Could it be due to indirect effects on reactivation as the DUB plays many important roles in virus replication beyond the proposed control of RQC.”*

Response: we have addressed this issue by searching for putative quadruplexes in the ORFs of the viral proteins included in our analysis, which returned at least one quadruplex in each tested protein (see supplementary Table S2). However, quadruplexes are not the only hindrance to translation elongation that may be found in viral ORFs. For example, long

polyproline repeats are found in EBNA2 and other latency proteins, and hydrophobic domains are found in many structural proteins. We have provided data showing that the vDUB does not affect the efficiency of virus reactivation, as confirmed by the similar expression of the BZLF1 transactivator in cells expressing catalytically active and mutant BPLF1 (see Figure S4). The capacity of the long G4 forming region EBNA1 and the homolog encoded by KSHV to stall translation was reported previously and confirmed by three independent groups (see Refs #65, #86, #87). Given our finding that BPLF1 promotes translation across stall-inducing mRNA sequences, it is reasonable to hypothesize that this mechanism may contribute to the efficient upregulation of EBNA1 in cells that express the active vDUB. Our data also suggest that this is not the only mechanism by which BPLF1 may promote translation since the activation of ISR would also enhance uORF and IRES-dependent translation initiation. Our conclusion that catalytically active BPLF1 enhances the translation of EBNA1 and other viral proteins, as opposed to other possible mechanisms, is based on the finding that: 1. it does not significantly affect the levels of transcription (see Figure 6C and S8B); 2. The effect is independent of the capacity of the vDUB to inhibit proteasomal degradation since the expression is not rescued by treating LCLs carrying the mutant virus with MG132 that inhibits proteasome and, to some extent, lysosome-dependent degradation (see Figure S8C, S8D). These are the defining features of a bona fide translational effect.

7. *"Figure 2C-D needs to measure mRNA levels to ensure BPFL1 is not simply stabilizing the mRNA"*

Response: see our response to point 3 c)

1st review minor comments.

1. *"In the introduction, the authors make several claims about why RQC might function during infection, but this jumps back and forth between IRESs and re-initiation, normal host shutoff versus potential roles for ISR. It also mentions that viruses have "challenging" RNAs and deplete tRNAs, but many of the references cited are not virus studies and the general review on the role of tRNAs in viral infection doesn't mention tRNA depletion resulting in translation issues. It would be helpful to be more specific in discussing the role of ZNF598-mediated RQC which largely operates on elongating ribosomes without mixing it with cap-independent initiation concepts, and provide more direct references supporting statements about viral systems such as tRNA depletion."*

Response: We appreciate the reviewer's concerns and acknowledge that our arguments needed to be more clearly formulated and supported by relevant references. In response, we have performed new experiments, extensively revised the text of the Introduction and Discussion, substituted inaccurate references with more relevant ones, and added new ones. With the support of new data on the activation of ZAK α , we have clarified our thoughts on the mechanism by which BPLF1 may activate the ISR and the possible effect of BPLF1 on both translation initiation via activation of cap-independent translation and translation elongation via inhibition of the RQC and the readthrough to stall inducing mRNAs. We believe that this extensive revision has significantly improved that manuscript, and we are grateful for the reviewer's comments.

2. *"Aspects of the Discussion should also be adjusted to be clearer and more accurate. There are bold claims made that are not substantiated, such as BPFL1 inactivates ZNF598 or counteracts LTN1-mediated degradation of substrates, which are never*

experimentally tested. There are recent papers that offer an explanation for the apparent contradiction in RQC activity during poxvirus infection. In general, the Discussion is quite open ended and doesn't present a coherent model. It mentions PKR activation and ISR, but talks about IRESs and alternative initiation etc. But what is the evidence the RQC process being studied here is involved, based on points above, and what's the overall model for how RQC might control translation of so many different viral mRNAs during EBV infection? Is it through q-quadruplexes in all these mRNAs or broader effects?"

Response: see above regarding our extensive revision of the text, which has made our arguments more precise and accurate. We have experimentally validated the effect of BPLF1 on the ZNF598-mediated ubiquitination of ribosomal proteins (see Figure 2a, 2C). We have added references that discuss the possible mechanisms by which poxviruses regulate the RQC (see Ref #79, #80). We are discussing the potential significance of the viral-induced inactivation of PKR mediate ISR versus GCN2-mediate activation of the ISR (see Discussion page 22). We are discussing a scenario where the inactivation of the RQC and the concomitant activation of a ZAK α -GCN2 mediated ISR may favor both translation initiation and elongation, which may affect the expression of many different viral proteins (manuscript R1, Discussion page 22-23). We appreciate that our language may have been, in some cases, too assertive and have done our best to provide a more balanced formulation of our thoughts.

3. *"The PKR inhibitor blot in 3C seems to suggest there is an effect as good as ISRIB".*

Response: we acknowledge that the blot did not clearly illustrate our conclusions based on the results of repeated experiments. We have now provided a new blot that better represents the average of several experiments (see Figures 4D and 4E)

4. *"BZFL1 should be measured in Figure S1 to ensure the BPFL1 mutant doesn't simply reduce expression of the reactivator."*

Response: The requested control was included in manuscript R1, Figure S6

5. *"In the abstract, perhaps qualify that the tegument protein is BPFL1 in line 3 as the name appears out of nowhere towards the end, with no context"*

Response: the abstract was modified as suggested

2nd review specific points

1. *"Although there is not a lot of new data added, with these revisions the authors convincingly show that there is an effect of BPLF1's DUB activity on eIF2 α phosphorylation and induction of a stress responses that involves GCN2 and ATF4. Moreover, this appears to contribute to viral protein production. In addition, BPLF1 can counteract the effects of ZF598 expression on RPS10/20 ubiquitination and affects readthrough of stall reporters. However, BPLF1 interacts with an enormous number of different proteins and complexes and has broad effects on the cell, and whether these effects on RQC are direct or an indirect consequence of independent activation of GCN2 remains unclear. As the authors themselves note, the ISR and RQC influence each other's activities, so there may be no direct connections in the reporter assays used in most of the paper. As suggested previously, do things like treatment with ISRIB*

decrease RQC and reporter readthrough? Is it activation of stress responses and eIF2 that affect RQC, or is it the other way around and RQC is influencing GCN2-eIF2 activation?"

Response: The reviewer acknowledges that, in our first manuscript revision, we have provided convincing data that link BPLF1 to both the IRS's activation and the RQC's inhibition. We have also shown that BPLF1 induces the activation of ZAK α , which is known to activate the GCN2 kinase that phosphorylates eIF2 α , and further confirmed that GCN2 is the kinase that phosphorylates eIF2 α in BPLF1 expressing cells. These findings support our proposed scenario where inhibition of the RQC promotes persistent ribosome stalling, which promotes the activation of ZAK α and the subsequent activation of the ISR. The reviewer points out that we have not formally excluded that the opposite may occur, i.e., BPLF1 may first activate the ISR, which, via inhibition of general translation, may inhibit the RQC. We want to point out that our findings that PKR and PERK are not involved in the phosphorylation of eIF2 α in BPLF1 expression cells (see Figure 4D, 4E) already exclude many possible pathways by which BPLF1 could activate the ISR, independently of any effect on the RQC, including for example the activation of ER stress or IFN responses. We are sorry for not understanding that the reviewer specifically requested that we test the impact of ISRIB on the regulation of the RQC by BPLF1. As discussed in our previous rebuttal, we are hesitant about the rationale and interpretation of this experiment because we expect that, by restoring cap-dependent translation, ISRIB will cause a translational overload that may activate the RQC. This may exacerbate the stalling of ribosomes induced by ANS treatment or poly(A) sequences but will not affect the activity of the vDUB. We have now performed the suggested experiment and included the results in Figure S5. As predicted, treatment with ISRIB did not affect the capacity of the vDUB to inhibit 40S ribosome ubiquitination in ANS-treated cells and did not alter the rescue of RQC substrates, which supports our model where activation of the ISR is not required for the capacity of BPLF1 to inhibit the RQC.

2. *"The biggest issue with the initial and revised submissions is that they only briefly test the role of BPLF1's DUB activity and GCN2 during infection, but draw sweeping conclusions from limited experiments. Again, it is relatively convincing that GCN2 is important for production of viral proteins, but arguments that DUB activity is regulating viral translation based on the argument the proteins are very stable are not convincing, particularly when studying a ubiquitination regulator with broad functions. Worse still, the authors loosely identify putative GC-rich sequences and build a model that evokes both stalling on these sequences as the driver of translational arrest through RQC and ISR, yet initiation on these same mRNAs occurs using uORFs and IRESs. None of these RNA elements are tested and the model is pure speculation and ultimately, quite confused. Overall, the core of the paper is based on transfection approaches that study a viral protein in isolation and outside the context of infection, but there is very little work done to then relate this to what the authors propose is happening during infection. What makes this even more speculative is the comparisons with other viral DUBs which, as noted before, raises concerns that DUBs that do regulate RQC are doing so largely through indirect effects on stress pathways. Aren't all herpesviruses GC-rich? So, why would only some use this strategy but not others? This data is simply added without any real exploration or context."*

a) **Rev:** *"It is relatively convincing that GCN2 is important for production of viral proteins, but arguments that DUB activity is regulating viral translation based on the argument the proteins are very stable are not convincing, particularly when studying a ubiquitination regulator with broad functions."*

Resp: We base our conclusion on the effect of BPLF1 on translation on three findings: i. the vDUB does not affect the efficiency of virus reactivation as assessed by the expression of BZLF1 (see Figure S6), ii. the vDUB does not significantly affect the transcription of viral mRNAs (see Figures 6C and S8B), iii. the enhanced expression of viral proteins in cells expressing the active vDUB is not explained by a general stabilizing effect on protein turnover. The resistance of EBNA1 to proteasomal degradation was amply demonstrated and is confirmed for other viral proteins in the manuscript (see Figures S8C and S8D). In this revision, we complement these data by showing that MG132 does not rescue the upregulation of EBNA in LCL carrying the mutant BPLF1 (see Figure S7). These findings support our conclusion that BPLF1 enhances the expression of viral proteins at the translation level rather than transcriptionally or post-transcriptionally.

b) Rev: "Worse still, the authors loosely identify putative GC-rich sequences and build a model that evokes both stalling on these sequences as the driver of translational arrest through RQC and ISR, yet initiation on these same mRNAs occurs using uORFs and IRESs. None of these RNA elements are tested and the model is pure speculation and ultimately, quite confused."

Resp: It appears that the reviewer may have misunderstood our model. Our findings show that the vDUB inhibits the RQC response by counteracting the ZNF598-mediated ubiquitination of 40S ribosome proteins. As reported in the literature and confirmed by our data, this will have two effects: 1. failure of the RQC-mediated disassembly of the stalled ribosome and degradation of the arrested translation products, which will allow, after pausing, the restart of translation and readthrough of stall-inducing sequences; 2. the prolonged stalling and ribosome collision induced by the inactivation of the RQC will promote the activation of ZAK α , which will activate the ISR via the phosphorylation of GCN2. While inhibition of the RQC will promote translation elongation, activation of the ISR will promote the translation initiated via uORF and IRES mRNA domains. The two effects are not contradictory, and they may synergistically promote the translation of viral mRNAs, although their relative contribution may vary depending on the features of the individual mRNA. The capacity of G4 sequences to inhibit translation is well documented, also in the context of herpesvirus proteins, and so is the involvement of uORF and IRES sequences in the initiation of translation upon activation of the ISR. We believe that our model, presented in Figure 8, is well grounded in the current findings and the available literature. We are not proposing that the very same mRNA simultaneously triggers the RQC and is then translated due to activation of the ISR, although this is entirely possible. Many features of viral mRNAs may promote ribosome stalling, including polyproline (several EBV nuclear antigens) or hydrophobic domain coding regions (several structural proteins). We have further clarified our thoughts on this issue in the Discussion page 22-23.

Rev: "Overall, the core of the paper is based on transfection approaches that study a viral protein in isolation and outside the context of infection, but there is very little work done to then relate this to what the authors propose is happening during infection"

Resp: We respectfully disagree with this comment of the review. It is correct that our findings illustrating the effect of BPLF1 on inhibition of the RQC and activation of the ISR were based on reporter systems and transfection approaches. Still, both findings

were validated in a relevant cell model of viral infection, i.e., EBV-transformed LCLs that express physiological levels of catalytically active and inactive BPLF1. We show that the active vDUB promotes the translation of EBNA1 that contains a well-documented stall-inducing sequence (see Figure 6B, 6C), and that the translation of EBNA1 is strongly enhanced by the GCN2-dependent activation of the ISR (see Figure 6D). These novel findings bring us as close as possible to demonstrating that BPLF1 regulates the RQC and ISR in EBV infected cells under physiological expression levels. We should also add the LCLs are very resistant to transfection and LCLs carrying recombinant EBV already express GFP and several selection markers, which precludes validation experiments such as transfection of a bona fide stalling reporter.

c) ***“What makes this even more speculative is the comparisons with other viral DUBs which, as noted before, raises concerns that DUBs that do regulate RQC are doing so largely through indirect effects on stress pathways. Aren’t all herpesviruses GC-rich? So, why would only some use this strategy but not others? This data is simply added without any real exploration or context.”***

Resp: We have already commented on this point in our previous rebuttal. We are further discussing the possible significance of the different functions of the vDUBs in the revised Discussion (Discussion page 24). The alpha-herpesviruses differ from the beta- and gamma-herpesviruses in both host range and replicative behavior. The vDUB domains are also quite different in amino acid sequence and surface charge, likely resulting in different putative substrates. That the alpha-herpesviruses use different strategies to interfere with ubiquitin/UbL-regulated processes is also supported by their expression of a viral ubiquitin ligase that is not conserved in the beta- and gamma-herpesviruses. Thus, the failure of the HSV1 vDUB to interfere with the RQC and ISR through the mechanism employed by the beta and gamma virus does not invalidate our findings.

3. *“Neither of these key points, nor many others, were actually addressed in the revisions. Indeed, the authors used summarization of the prior comments in a way that seemed to write off and ignore many important issues. This includes several important controls to establish baseline effects that also seem applicable to new data added. Overall, rather than boiling reviewer points down to a one-liner that often omits the main point, the authors should have addressed each concern directly in their rebuttal and revisions.”*

Response: as stated at the beginning of this rebuttal, we are very sorry if our attempt to conceptualize our response to the first review gave the impression of negligence. We hope our detailed response to the first and second set of comments will reassure and satisfy the reviewer.

4. *“As stated previously, in most experiments, the GFP control is only tested in the control condition but it would be important to rule out that BPFL1 doesn’t simply increase expression of GFP in general, rather than just the readthrough products. For example, where a control is included, namely the KO control shown in Figure 3D, both GFP and RFP seem increased by BPFL1. This could be hinting at indirect effects of this DUB on protein stability which may dramatically influence reporter activity”.*

Response: We have individually addressed the reviewer's comments on the lack of controls in our rebuttal to the first review. Several controls were included in the first revision, and, in some cases, we have explained our arguments for not including the suggested controls. Examples of the effect of BPLF1 on the expression of GFP are shown in Figures 3D and S5. As for the comment on the different expression of GFP in the KO transfected cells shown in Figure 3D. A slight difference was indeed observed in the experiment shown in the figure. This is why the densitometry values were normalized to the expression level of GFP, which is our reference for transfection and translation efficiency. As the reviewer may have noticed, similar levels of GFP are detected in all cells transfected with the K20 reporter, independently of BPLF1 expression (manuscript R2, Figure 3D), which excludes a systematic effect of the vDUB on the expression of the reporter.

5. *"The authors never really address how such large effects are seen with such low transfection efficiency, particularly in co-transfection assays. The images show 2/11 cells expressing, which is hard to reconcile with large effects on overall translation in the culture, and the co-transfection efficiency is unclear, yet FACS show dramatic effects in most expressing cells. I don't understand the gating argument being made for this as it doesn't address co-transfection efficiency."*

Response: We commented on transfection efficiency in responding to the first review. We have now modified the Material and Methods description of the transfection procedure by mentioning that optimizations were performed for each cell line (M&M page 24). We acknowledge that the images selected to document the effect of BPLF1 on puromycin incorporation may give the impression of low transfection efficiency. A new, more representative set of images from the same experiment is now shown in Figure 5C upper panels.

6. *"Why does ANS treatment not induce eIF2 phosphorylation as it should in control samples in Figure S1, despite activation of ZAK as expected? This may suggest that the authors are using the wrong concentration of ANS (see Wu et al, Cell, 2020)".*

Response: In this figure, we aim to confirm the finding reported in the literature that inactivation of RQC by ZNF598-KO promotes the autophosphorylation of ZAK α and activation of the ISR. The reviewer correctly notices that the phosphorylation of eIF2 α is comparatively lower in RQC-proficient cells. It is, however, not absent, as confirmed by the low but detectable upregulation of ATF4. We have chosen our doses of ANS based on the data reported in the paper mentioned by the reviewer, and we have also confirmed the dosage in titration experiments. This is now mentioned in the Results section, and a representative titration experiment is included in the revised manuscript (see Results page 7 and Figure S1)

7. *"The new data showing ZAK band-shifting in Figure 4A is not convincing as the shifts seem very small, other than in the ANS-treated controls, while BPLF1 and GAPDH all show different band-shifts under different conditions. One could argue the mutant also causes some level of shift, but again there is no untreated control for this sample to set the baseline."*

Response: the reviewer correctly notices that the size shift induced by both ANS treatment and BPLF1 expression is small. This aligns with published data (see for example Ref # 56). We have now repeated the experiment and treated the lysates with phosphatase rather than omitting the phosphatase inhibitors from the lysis buffer (see Figure 4A). The new results illustrate more clearly the size shift. The reviewer also notices that a small shift is observed

also in cells expressing BPLF1^{C61A}. Several controls were included in the new figure to account for nonspecific effects associated with transfection stress. We are discussing the significance of the different migration shifts observed in ANS and BPLF1 expressing cells in Results page 12-13. In the new figure, the effect of BPLF1 on the activation of ZAK α is substantiated by the finding that ZAK α substrates, p38, JNK are also phosphorylated.

8. *"In Figure 6B, I understand that there is no antibody available but there is a major assumption that BPLF1 protein is made in the reactivation systems. Since the authors went to the trouble of making a viral mutant it seems odd that they wouldn't include a Flag tag to confirm protein production, and at least similar levels between wildtype and mutant forms. I also could not find any details of how the mutant or cell lines were made, or any reference to their construction to refer to"*

Response: The reviewer missed the references describing the production of the recombinant viruses and immortalized LCLs (see Ref # 40 and references thereof, for production of recombinant EBV, and reference #69 for the establishment of LCLs carrying recombinant EBV expressing wild-type and mutant BPLF1). The mutant virus was produced by our collaborators by mutagenesis of the wild-type virus that does not carry FLAG-tagged BPLF1. Since the BPLF1 mRNA is more than 9000 bp and the recombinant virus is more than 100 kb, inserting a tag is not trivial. Clearly, the lack of a suitable antibody detecting the endogenous protein in induced LCLs is a limitation of our study. Still, the equal expression of the mRNAs and the functional data obtained with the recombinant vDUB domains strongly support our model.

Reviewer #2

We appreciate the constructive comments of the reviewer that have helped us to improve the manuscript significantly. We will here recapitulate our point-by-point response to the first review and address the remaining issues highlighted in the second review.

1st review major comments.

1. *"I wonder, if BLF1 (vDUB) interferes with RQC and ISR directly on the ribosome? I would strongly assume, that it binds to stalled, most likely collided ribosomes and there encounters the players that are initiating downstream quality control processes. Thus, here, I would be very interested so see, if FLAG-tagged BPLF1 co-migrates with ribosomes/polysomes on a sucrose gradient (before and after ANS treatment). RNase I (or S7 nuclease) treatment would even answer the question if BPLF1 binds directly or indirectly to 80S."*

Response: the reviewer suggests a very relevant experiment. We are currently setting up the methodology and hope to be able to perform this analysis in the continuation of our studies. However, we believe that our findings that BPLF1 is found in protein complexes that contain ribosome proteins and ZNF598, together with the effect on the ubiquitination of 40S protein subunits in ANS-treated cells, places the DUB in close vicinity if not in direct contact with stalled ribosomes. In the revised manuscript, we have at least partially closed this gap by producing a ZNF598-KO cells line where we could confirm that expression of the vDUB directly impacts ZNF598-dependent ubiquitination (manuscript R1, Figure 2C)

2. *"One central player that acts at the interface of RQC and ISR on colliding ribosomes (especially when considering prolonged stalls) is ZAK α . While I appreciate, that*

ZAK α is not found in the MS analysis of the BLF1 interactome, it still doesn't exclude that ZAK α may play a role in the described pathway. Therefore, I would be happy if the authors could address the question, if ZAK α can influence BLF1 (vDUB) activity by any experiment of their choice."

Response: we are very grateful for this insightful suggestion. In the revised manuscript, we show that BPLF1 is found in complexes containing ZAK α (manuscript R1, Figure 1D) and that ZAK α is phosphorylated in cells expressing the active vDUB (manuscript R1, Figure 4A). These findings are consistent with our proposed model where, by inhibiting the RQC, the vDUB promotes prolonged ribosome stalling and collision, which, via autophosphorylation of ZAK α and activation of GCN2, triggers the ISR.

3. *"Introduction: With respect to ribosome collision and RQC, the intro seems a little imprecise or out of date. For example: 1.) collided trisomes were analyzed in Matsuo et al., NSMB 2020 (ref 12), not in Ikeuchi et al., (ref 3; this one is for disomes). 2.) for ubiquitination of ribosomal proteins, please add ref Narita et al. Nat Comms 2023 (PMID: 36302773). This one shows that uS10, not eS10 poly-ubiquitination (K63-linked) is required for subsequent splitting by the ASCC. It also shows a human disome structure. 3.) When reporting about the interdependence between ribosome collision, RQC and ISR, the authors should address kinase ZAK α . For example, in Wu et al (cited in ref 21) was shown, that under intermediate doses of anisomycin, eIF2 α phosphorylation is reduced in ZAK KO cells. I consider this quite relevant also for this study, since authors show that vDUB decreases 40S protein ubiquitination exactly after anisomycin treatment. 4.) on page 4 in think RSP2 and RSP3 should be RPS2 and RPS3."*

Response: we are again very grateful for the valuable remarks and suggestions. We have extensively revised the introduction, corrected several printing mistakes, and added the suggested references.

4. *"Results: Related to Figure 1A: Although not listed, I wonder if ZAK was also a hit in the BPLF1 interactome. Moreover, since UFL1 was found, what about the other two components of the E3 ligase, CDK5RAP3 and DDRGK1? There is some recent work that the UFMylation machinery plays an important role in ER-RQC (e.g., PMID: 3694557). The fact that these proteins are found together with components of protein targeting and translocation to the ER hints at a function of BPLF1 on ribosomes stalled at the ER. Can the authors comment on an enrichment of BPLF1 on ER membranes after anisomycin-mediated ribosome stalling?"*

Response: we have performed the suggested experiment and found that ZAK α is indeed present in protein complexes containing BPLF1 (manuscript R1, Figure 1D) and is phosphorylated in BPLF1 expressing cells (manuscript R1, Figure 4A). We have commented on the possible involvement of BPLF1 in the regulation of the ER-RQC in manuscript R1, Discussion page 18.

5. *"Results: eIF5A is not part of the pre-initiation complex (see comment below)"*

Response: we have corrected the text and added the correct reference

6. *"Results: Typo in legend: Western bots (in D) Related to Figure 2:"*

Response: we have corrected the mistake

7. *“Results: The Western Blot signals for Ub-RPS3 are rather weak and intensities are hard to compare (could be moved to Supplementary), whereas the signal for Ub-RPS10 is quite clear and effects for BPLF1 are convincing. Yet, I wonder why the authors didn’t check of RPS20 (uS10). Especially, according to a recent study (Narita et al. NSMB 2023), uS10 should be the main target of ZNF598 leading to disassembly of collided ribosomes. Could the authors add this experiment? A comment on weak RPS3 signals: This makes sense in a way, since at least in yeast, RPS3-ubiquitination (by RNF10 homolog Mag2 and Fap1) is rather required to resolve stalled monosomes (e.g., after failed initiation), and less after collision. It is quite likely, that this is similar in humans. Please also check Li et al, Mol. Cell 2022 for Discussion (PMID: 36113412).”*

Response: as suggested, we have performed a new set of experiments that confirm the capacity of BPLF1 to inhibit the ubiquitination of both RPS10 and RPS20 (**manuscript R1, Figure 2A, 2B**). The ability of BPLF1 to inhibit the ubiquitination of RPS3 is very interesting, but we agree that it is not directly relevant to the current manuscript. The data were omitted in the revised manuscript.

8. *“Results: Related to Figure 3: I wonder if the increased activation of eIF2 α -phosphorylation by BPLF is in any way correlated with ZAK α activation. Would it be possible to repeat the experiment shown in Fig. 3A and 3C (with GCN2i) in ZAK α KO cells? (Or any other experiment that addresses the question, if ZAK α plays a role for BPLF activity?). I would also be interested in what happens to eIF2 α -P, CHOP and ATF4 expression (Fig. 3A) after ANS addition.”*

Response: see above regarding the inclusion of data on ZAK α phosphorylation. The effect of ANS on the expression of eIF2 α and ATF4 is shown in the new **supplementary Figure S1, S2**. As expected, the effect is weak in RQC-proficient cells and much more potent in cells lacking ZNF598 (**Figure S2**), consistent with the strong effect observed in cells expressing the active vDUB.

9. *“Related to Discussion: On page 15, last paragraph, the authors discuss the*

l
Response: we have extensively revised the Discussion and, as suggested, omitted the data on RSP3 ubiquitination.

10. *“Related to Discussion: Would it be possible to add a cartoon or graphic*

Response: a cartoon illustrating our models for the regulation of the RQC and ISR by BPLF1 and the possible consequences for viral gene expression is now included (**manuscript R1, Figure 8**)

11. *”*

Response: we have revised the table to the best of our knowledge

”

2nd review specific points

1. *“Compared to the first submission, the revised manuscript is significantly*

”

Response: we again thank the reviewer for a very valuable suggestion. Although the experiment is not directly relevant to the regulation of the RQC and ISR by the vDUB, we have performed the investigation, and the results are now shown in **manuscript R2 Figure 4A**. In line with the finding reported in the literature, we find that the activation of ZAK α induced by BPLF1 also correlates with phosphorylation of p38 and JNK.

2. *"Another interesting question for me that was not clarified is, if BPLF1 is indeed*

Response: we again appreciate this comment and recognize the relevance of the suggested experiment. Regretfully, we are still unable to include the data. However, as discussed in our previous rebuttal, we believe that direct proof of association of the vDUB with collided ribosomes is not essential for a correct interpretation of our findings since both the ubiquitination of 40S ribosome protein by ZNF598 and the autophosphorylation of ZAK α occur on collided ribosomes.

3. *"Minor comments: 1.) In case no experimental data on RSR are available, please*

Response: see above for the new experiments regarding the activation of the RSR pathway. This aspect of the ribosome stress response is now mentioned in the Introduction (**manuscript R2, Introduction page 4**). We have corrected the text and printing mistakes and again revised the supplementary Table S1.

*l
d*

Reviewer # 3

The reviewer considers our manuscript revision satisfactory and has no further comments. We will briefly summarize our response to the critique forwarded in the first review.

1st revision major comments

e

1. *"*

Response: as discussed in our rebuttal to the comments of reviewer #1, we are not claiming that BPLF1 directly interacts with ZNF598, but the vDUB is present in protein complexes that contain the RQC ligases. Furthermore, in the revised manuscript, we confirm that BPLF1 directly affects the ubiquitination of RPS10 and RPS20 upon reconstitution of ZNF598 in ZNF598-KO cells (**manuscript R1, Figure 2C**). To our knowledge, direct interaction with the ligase is not a prerequisite for deubiquitination of the substrate. While interesting, pool-down assays of recombinant proteins aiming to identify the primary interactor that recruits the vDUB to the complex are not within the scope of the manuscript.

O

"Fig 2A: The role of BPLF1 in the deubiquitylation of RPS10 and RPS3 remains unclear. The

Response: we addressed this concern by producing a ZNF598-KO cell line where we found that BPLF1 inhibits the ubiquitination of RPS10 and RPS20 induced by overexpression of the ligase in the absence of ANS treatment (**manuscript R1, Figure 2C**). In vitro deubiquitination assays will not be informative because BPLF1 is a very potent DUB that, taken outside of the cellular context, will most likely deubiquitinate any substrate.

*r
N
"
t
i
n
g
"*

Response: see our response to point #1. Our new data obtained with the ZNF598-KO cells conclusively demonstrate the capacity of BPLF1 to counteract the activity of ZNF598. Since ZNF598 only ubiquitinates 40S proteins on collided ribosomes, the suggested experiment will not add significantly to our findings.

”

Response: in the revised manuscript, we provide new data confirming the rescue and readthrough of a stalling reporter in ZNF-KO cells (manuscript R1, Figure S3). Further validation of the pathway in LTN1-KO cells will not add significantly to this conclusion.

Response: We believe that the role of BPLF1 in the activation of the ISR is adequately addressed by the new experiments showing that BPLF1 coprecipitates with ZAK α -containing protein complexes and promotes ZAK α autophosphorylation (manuscript R1, Figure 4A). ZAK α activated on collided ribosomes phosphorylates the GCN2 kinase that we show to be involved in activating the ISR in BPLF1-expressing cells. We believe that these findings strongly support our model that connects the inhibition of the RQC by the vDUB with the activation of the ISR and the enhanced translation of viral proteins in BPLF1-expressing cells (manuscript R1, Figure 8)

REVIEWERS' COMMENTS

Reviewer #1 (Remarks to the Author):

Additional data and responses from the authors address my main concerns, and I believe that they now provide convincing evidence that BPFL1 can counteract ZNF598-mediated activation of RQC in polyA-based model systems and contributes to viral protein production during infection. I do still think that the authors could have tested more natural viral RNA elements and provided better evidence that these events actually occur during infection (e.g. show ZAK/GCN2/eIF2 activation), particularly as for example other viral proteins are reported activate ATF4 through PERK (e.g. doi: 10.1182/blood-2007-07-100032, doi: 10.1128/jvi.78.4.1657-1664.2004), but the findings are thought provoking and novel, the argument that G4 quadruplexes likely also cause stalling is reasonable, and time will tell if their more complex G4/uORF/IRES-GCN2/eIF2 model is correct.

Reviewer #2 (Remarks to the Author):

The authors also addressed my last remaining issues from the second revision and I have no further comments.

Point-by-Point Response

Reviewer #1

" Additional data and responses from the authors address my main concerns, and I believe that they now provide convincing evidence that BPFL1 can counteract ZNF598-mediated activation of RQC in polyA-based model systems and contributes to viral protein production during infection. I do still think that the authors could have tested more natural viral RNA elements and provided better evidence that these events actually occur during infection (e.g. show ZAK/GCN2/eIF2 activation), particularly as for example other viral proteins are reported activate ATF4 through PERK (e.g. doi: 10.1182/blood-2007-07-100032, doi: 10.1128/jvi.78.4.1657-1664.2004), but the findings are thought provoking and novel, the argument that G4 quadruplexes likely also cause stalling is reasonable, and time will tell if their more complex G4/uORF/IRES-GCN2/eIF2 model is correct."

Resp. We appreciate the positive response of the reviewer, who acknowledges the novelty of our findings. We are aware of the studies on the triggering of ER stress responses by the EBV latent membrane protein LMP1 mentioned by the reviewer. It should be noted that, unlike BPLF1, LMP1 is a membrane protein that is found both at the ER and outer cell membrane and is expressed during latency and productive infection. The toxic effects of LMP1 overexpression are well documented and may be ascribed to a non-specific disturbance of the ER homeostasis caused by the abundant expression of a foreign membrane protein. We share the view that additional work will be required to fully validate our model in the context of viral infection. In spite of the technical complexity of working with primary EBV-infected cells, as opposed to tumor cell lines where additional changes may be selected in vivo and during long-term in vitro culture, we are committed to continuing our investigation.

Reviewer #2

"The authors also addressed my last remaining issues from the second revision and I have no further comments."

Resp. We thank the reviewer for the positive response and are particularly grateful for the constructive criticism and valuable suggestions throughout the review process.